# Biodegradation of Polymers: Stages, Measurement, Standards and Prospects

**Rafael Resende Assis Silva** [1,*], **Clara Suprani Marques** [2], **Tarsila Rodrigues Arruda** [2], **Samiris Cocco Teixeira** [2] and **Taíla Veloso de Oliveira** [2]

1    Department of Materials Science and Engineering, Federal University of São Carlos, São Carlos 13565-905, Brazil
2    Food Technology Department, Federal University of Viçosa, Viçosa 36570-900, Brazil
*    Correspondence: rafaelras@estudante.ufscar.br; Tel.: +55-(37)-99131-3471

**Abstract:** Nowadays, sustainable and biodegradable bioplastics are gaining significant attention due to resource depletion and plastic pollution. An increasing number of environmentally friendly plastics are being introduced to the market with the aim of addressing these concerns. However, many final products still contain additives or mix non-biodegradable polymers to ensure minimum performance, which often undermines their ecological footprint. Moreover, there is a lack of knowledge about all stages of biodegradation and their accuracy in classifying products as biodegradable. Therefore, this review provides an overview of biodegradable polymers, elucidating the steps and mechanisms of polymer biodegradation. We also caution readers about the growing marketing practice of "greenwashing" where companies or organizations adopt green marketing strategies to label products with more environmental benefits than they have. Furthermore, we present the main standards for evaluating biodegradation, tools, and tests capable of measuring the biodegradation process. Finally, we suggest strategies and perspectives involving concepts of recycling and the circularity of polymers to make them more environmentally friendly and sustainable. After all, "throwing away" plastics should not be an option because there is no outside when there is only one planet.

**Keywords:** biodegradation; biodegradable plastics; biodegradable polymers; biodegradation methods; biodeterioration; greenwashing; circular economy

## 1. Introduction and Definitions

Nowadays, plastic materials are an essential part of the society. Due to their versatility and relatively low cost compared to other available materials, plastics have been employed in several outcomes, including a wide range of industries, commerce, and households. It is safe to say that plastics are ubiquitous. Although global plastics production slightly decreased in 2020, the COVID-19 pandemic triggered an increase in plastics demand related to medical items, personal protective equipment, and the enhancement of delivery systems [1].

However, their incorrect waste disposal is a serious problem that humanity currently faces. It leads to adverse environmental impacts and can cause significant health issues. One of the main causes is the mass production and increased use of plastics that are still largely produced from fossil sources and which exhibit non-biodegradable behavior [2]. In general, proper recycling is the most effective solution for plastic waste management. However, the high heterogeneity (mixture of plastics and the miscellaneous additives conventionally employed in plastics production, such as plasticizers, antioxidants, and stabilizers), recalcitrance, and cross-contamination of plastic wastes with organic matter and inert materials typically make thermochemical processing arduous, costly, and energy-intensive. Other alternatives may include waste incineration, which potentially negatively impacts the environment and human health, with the great production of toxic

gases (e.g., furans, dioxins, polychlorinated biphenyls, and mercury) [1,3]. In this view, bioplastics have been developed as an alternative to conventional plastics [4,5]. But what is the concept of bioplastics? Bioplastics are still ill-defined. In fact, a bioplastics refers to a polymer that is produced from natural sources or renewable resources and then utilized in the production of commercial products, e.g., starch, and cellulose. They may comprise materials that are supposed to degrade naturally (biodegradable, such as polybutylene succinate "PBS" and polycaprolactone "PCL"), materials made from renewable feedstocks (bio-based, such as bio-polyethylene "Bio-PE"), or both (such as polylactic acid, "PLA", chitosan, proteins, cellulose derivatives) [6,7]. Therefore, the use of bioplastics in substitution for traditional (fossil-sourced) plastics for single use can be an interesting point of view since they theoretically present similar favorable properties as their petroleum-based counterparts, such as cheap, lightweight, flexible. Additionally, bioplastics offer the advantages of being derived from renewable resources, contributing to circularity, reducing carbon footprint, and, in some cases, exhibiting biodegradability within a reasonable timeframe. However, there is still a lack of scientific evidence concerning the real biodegradability and sustainability of these novel plastics, especially when structured in the form of composites and/or blends [7].The term "biodegradation" can be misleading, as all plastics, including conventional ones, will eventually undergo biodegradation. However, the length of time required for this process is a crucial factor, ranging from a few days to thousands of years. The IUPAC definition characterizes biodegradation as the "breakdown of a substance catalyzed by enzymes in vitro or in vivo" [8]. For the purposes of hazard assessment, biodegradation can be classified as (i) primary, which involves altering the chemical structure of a substance resulting in the loss of a specific property; (ii) environmentally acceptable, referring to biodegradation to the extent that undesirable properties of the compound are removed. This often corresponds to primary biodegradation, but it depends on the circumstances under which the products are discharged into the environment; and (iii) ultimate, which involves complete compound breakdown to fully oxidized or reduced simple molecules (such as carbon dioxide/methane, nitrate/ammonium, and water). It should be noted that biodegradation products can be more harmful than the substance being degraded. This process can occur in the presence (aerobic) or absence (anaerobic) of oxygen. Figure 1 presents a general scheme of the biodegradation process.

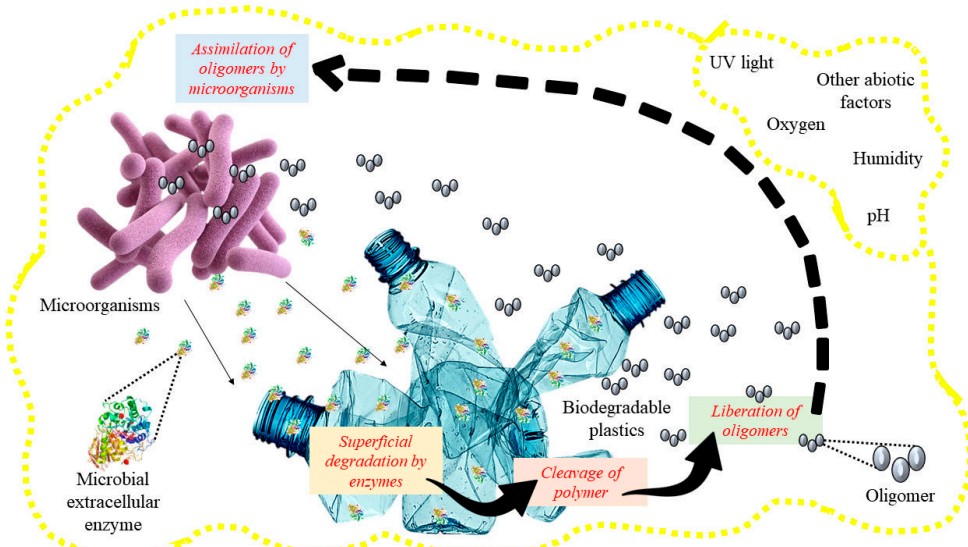

**Figure 1.** Generic representation of the biodegradation process.

In this context, biodegradable plastics were defined by the American Society for Testing and Materials (ASTM D883-17) [9] as "degradable plastic in which the degradation results from the action of naturally occurring microorganisms such as bacteria, fungi, and algae. Biodegradable polymers can also be classified as a materials designed to perform

for a finite period, undergoing a controlled degradation process into compounds that can be metabolized by microorganisms [2,10]. They can also be described as materials whose chemical and physical properties deteriorate and completely decompose when exposed to microorganisms and aerobic and anaerobic processes [11]. According to IUPAC [8], these materials are "polymers susceptible to degradation by biological activity (i.e., their biodegradation proceeds not only by the catalytic activity of enzymes, but also by a wide variety of biological activities), with the degradation accompanied by a lowering of their molar mass".

To ensure fairness in meaning, we define biodegradation as the degradation of materials, at least 90% by mass, into low molar mass products, such as water, carbon dioxide, and biomass, followed by assimilation by naturally available microorganisms under normal environmental conditions in a reasonably considerable time (~6 months). Similar to conventional plastic materials, bioplastics also require specific additives for their production, especially those produced from natural sources (e.g., cellulose, starch, proteins) or present in microbial production (e.g., PLA), since they exhibit specific physical properties, such as thermal resistance and barrier properties, that limit diverse applications [7,12]. These additives can migrate from the plastic or not be degraded, potentially posing toxicity risks to terrestrial and aquatic ecosystems. Therefore, it is extremely important to assess the biodegradability of each bioplastic produced, considering their particularities and the disposal system, as bioplastic is not necessarily synonymous with biodegradable plastic. Given this information, this review provides a comprehensive overview of the biodegradability of polymers. The following sections will present essential topics on the stages, measurement procedures, and standardization of plastics biodegradation, which will help to understand the prospects and perspectives of the usage, disposal, and environmental impacts of plastic materials.

## 2. Biodegradable Polymers and Stages of Biodegradation

### 2.1. Mechanisms of Degradation

Polymer degradation refers to any chemical, physical, or biochemical reaction that involves breaking covalent bonds in the backbone of the polymer, resulting in an irreversible change in its properties due to alterations in the chemical structure and the reduction of molecular weight. The breaking of primary chemical bonds in the main or side chain generates reactive species (free radicals) that are responsible for propagating the degradation process of the polymeric artifact. The initiation of the polymer degradation process is catalyzed by abiotic factors, e.g., heat, light, radiation, humidity, pH of the medium, mechanical stress, and chemical attack. These forms of initiation require activation energy for breaking chemical bonds in the polymer, with the binding energy varying according to the atoms' connection, i.e., they can have ionic, coordinate, metallic, or covalent primary bonds. Generally, the types of bonds in organic polymers are covalent and usually involve short distances and high energies (1.5 Å and 100 K/mol) [13]. The main covalent bonds found in organic polymers, their binding energy, stability, and binding distance, are discussed in-depth by Canevarolo (2006) [13].

Chain scission or bond breaking occurs when the localized energy in this chemical bond is greater than the energy of the bond. When a more unstable bond is positioned in side groups or short branches, its breakage leads to (i) the loss of that side group or (ii) its modification by the insertion of new atoms (e.g., oxygen), resulting in polymer degradation. This type of degradation can occur both in the solid and molten states. The energy required for bond scission can be provided in different ways, such as heat (thermolysis), water (hydrolysis), oxygen (oxidation), chemistry (solvolysis), light (photolysis), gamma radiation (radiolysis), or shear (mechanical) or weathering (generally UV/ozone degradation), etc. Here, we will focus on the first three types of degradation since they present a higher occurrence of scissions in polymers.

2.1.1. Hydrolysis

Hydrolysis is a chemical decomposition process that involves breaking a bond by reacting with water molecules. The hydrolysis process is the most important for initiating the biodegradation of synthetic polymers, especially polyesters. The rate of hydrolytic degradation varies from a few hours to years, depending mainly on the degree of crystallinity, type of functional group, molecular weight, main skeletal structure, morphology, temperature, and pH of the medium. According to Lyu & Untereker (2009) [14], hydrolytic degradation is divided into three levels. The first level involves degradation at the molecular level, in which hydrolysis is controlled only by chemical reactivity. The second level is also molecular but is associated with molecular mobility and water–polymer interactions. The third level is the macroscopic one, where erosion and water diffusion reaction are the governing parameters for degradation.

Therefore, hydrolysis can cause biopolymers to degrade either through surface erosion or bulk erosion. During surface erosion, the outer layer of the polymer degrades first, while the inner material is degraded last. In contrast, bulk erosion occurs when water molecules quickly diffuse into the amorphous regions of the polymer, causing a rapid loss of strength and structural properties [12].

Hydrolysis occurs mainly in hygroscopic polymers and those with water-sensitive groups in the polymeric backbone. During hydrolysis, the polymer is always split into two components; otherwise, it will not be considered hydrolysis (hydro = water; lysis = breakdown). If the products are not ionized, one part gains a hydrogen atom (H+), and the other gains a hydroxyl group (OH-) from the broken water molecule. Figure 2 shows the hydrolysis rate ranking of the main polymers that undergo degradation when exposed to moisture, e.g., polyanhydrides, polyesters, polyethers, polyamides, polycarbonates, etc. Furthermore, it is shown that hydrolysis also depends on the polymer's polarity and degree of crystallinity. More hydrophobic polymers have a lower reaction rate because the water content in the polymer and the water permeability decrease with decreasing polymer polarity. Therefore, hydrolytic stability increases in the same order as hydrophobicity. In turn, an increase in crystalline phases in polymers inhibits the plasticization of the polymer by the water in these regions since the steric effect and strong intermolecular interactions impede water penetration in the ordered regions, i.e., crystalline.

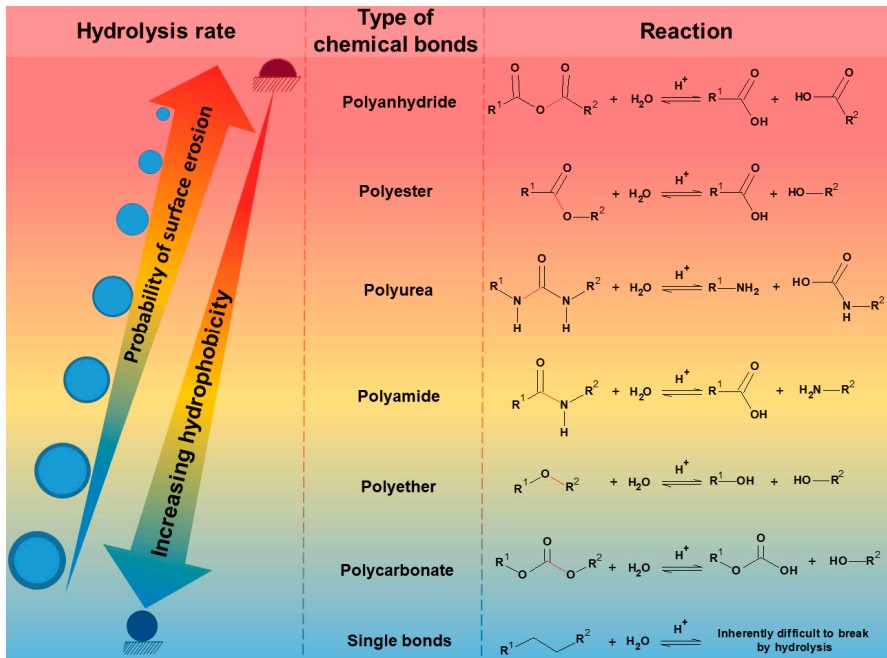

**Figure 2.** Ranking of the hydrolysis rate of the main polymers that suffer degradation when exposed to humidity.

In turn, if the polymers become ionized after separation, one part will receive two hydrogen atoms with a localized positive charge, while the other part will have an oxygen atom with a negative charge. For example, amino acids are released from protein chains by hydrolysis (Figure 3a). Silva et al. (2021) [15] showed that the absorption of water in the polymeric matrix could reduce both the temperature of decomposition of the polymers and act as a plasticizing effect, i.e., reducing the glass transition temperature of the polymers. The reduction of glass transition temperature (Tg) by water absorption is one of the most significant effects in modifying the properties of plastics, as water reduces intermolecular interactions between polymeric chains. As a result, plastics have reduced stiffness (Young's modulus), tensile strength, and decomposition temperature [15]. Therefore, the reduction in the performance of these properties and the increase in water vapor permeability, catalyzed by water as a plasticizer, are critical parameters that make the use of plastics for food packaging unfeasible.

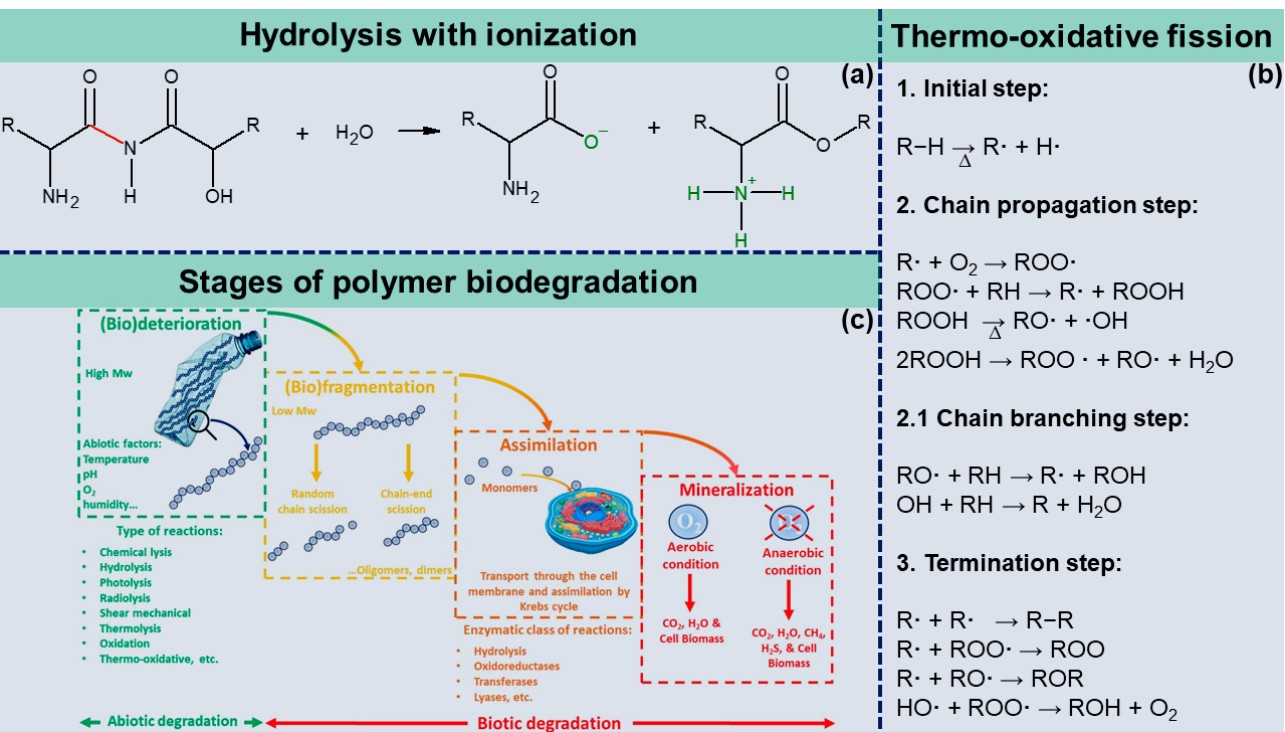

**Figure 3.** Ionization of an amino acid after hydrolysis (**a**); steps of thermo-oxidative degradation reactions (**b**); stages of polymer biodegradation (**c**).

Lyu & Untereker (2009) [14] demonstrated that water can dissolve in many polymers at a level of approximately 1% wt., which can increase the rate of degradation by hydrolysis. However, in other polymers, the rate of water penetration is much slower than the rate of reaction to break the polymer chains into soluble fragments, indicating that the polymer will degrade by surface erosion. Additionally, Silva et al. (2022) [16] showed that incorporating certain additives, such as LiCl, into polymers to create active antimicrobial packaging may have unintended side effects, such as increased water absorption due to the hygroscopic nature of the added filler. Therefore, a current challenge in the physicochemical and biodegradation of polymers is to investigate ways to synthesize or combine polymers that are water-resistant and can degrade rapidly at the end of their life cycle.

### 2.1.2. Thermolysis

Thermal decomposition, also known as thermolysis, is a chemical reaction in which a reacting substance decomposes into at least two new substances upon heating. In the case of polymers, thermal decomposition generates molecules and atoms that are different

from the precursor without the simultaneous involvement of other reagents such as oxygen. Since the heat received breaks the bonds of the molecules of the reactants, thermal decomposition is generally an endothermic process. If the chemical energy of the reactants is greater than that of the products, the decomposition reaction will be exothermic ($\Delta H$), indicating that the reactants are highly reactive and the products are stable. An exothermic decomposition reaction releases heat and may be accompanied by an explosion or another chemical reaction.

At this point, we must highlight a widespread error in the literature regarding the study of thermolysis through thermogravimetric analysis (TGA). The results should be conceptualized with the term "decomposition temperature" instead of "degradation temperature". Unfortunately, the latter term is treated as a synonym for the former, which is incorrect. Degradation temperature refers to the temperature at which loss of some function or property of the material being studied occurs. For example, protein denaturation, inactivation of active antimicrobial agents, change in color or transparency, and reduction in mechanical or barrier performance to gases. On the other hand, the decomposition temperature (TDT) should be used to discuss TGA results because it refers to the decomposition of the polymer into smaller molecules, constituent atoms, and/or the release of gases such as $CO_2$, $CH_4$, CO, etc. In this sense, the degradation temperature often occurs before the decomposition temperature because most properties and functions of materials are thermosensitive, and some properties depend on secondary (intermolecular) bonds that break at mild temperatures. Therefore, when the TGA detects mass loss, it is crucial to describe the event as thermal decomposition as it necessarily involves the breaking of primary bonds, confirming the occurrence of material thermolysis.

### 2.1.3. Oxidation and Thermo-Oxidative Fission

We would like to emphasize that degradation in the presence of oxygen not only leads to chain scission or the breaking of the $\sigma$ bond (R-C-C-R), but also breaks the $\pi$ bonds (R-C=C-R), resulting in the insertion of an oxygen atom (oxidation). Therefore, in oxidation reactions, a reduction in the average molar mass of the polymer is not necessarily observed, but a marked change in its physical and chemical properties, e.g., a color change of the material, may occur. Regardless of the atmosphere's composition, polymers will start to decompose if heated enough. However, thermal oxidation differs from thermal decomposition in that it generally catalyzes oxidation reactions culminating in material decomposition at milder temperatures.

Thermo-oxidative fission of polymers is a self-catalytic process that occurs in three stages: initiation, propagation, and termination. The oxygen molecule is considered a highly reactive chemical species, as it reacts quickly with any environmental free radicals. In the first step, heat-catalyzed degradation is initiated when polymer chains form radicals (R*) either by hydrogen abstraction or by homolytic scission of the C-C bond. Next, the propagation of degradation involves a series of intermediate reactions. The first intermediate step is the reaction of a free radical (R·) with an oxygen molecule ($O_2$), forming a peroxy radical (ROO·) that abstracts a hydrogen atom from another polymeric chain, producing a hydroperoxide (ROOH). Hydroperoxides are highly unstable; therefore, they decompose into two new free radicals, (RO·) + (·OH), which attack the polymer chain, abstracting labile hydrogens and introducing new radicals [17]. The thermo-oxidative reaction ends by recombining two radicals, forming stable products, or abstracting hydrogen or $\pi$ bonds. Figure 3b shows the thermo-oxidative degradation reactions elucidated above.

### 2.2. Abiotic and Biotic Degradation

The degradation process of a polymer depends on its intrinsic properties and the extrinsic conditions to which it is exposed, such as the biodiversity and occurrence of microorganisms, which vary locally. Therefore, the degradation of materials can generally be classified as abiotic (heat, radiation, oxygen, humidity, solvents/chemicals) or biotic (bacteria, fungi, algae). Abiotic degradation is usually the first stage after the end of

the useful life of the plastic, during which physical and chemical changes occur, but not biological actions, resulting in the modification of at least one property or characteristic of the material. Some of these alterations are visible to the naked eye, such as changes in color, dimensions, cracks, and weight, while others require tools for characterization, such as mechanical and rheological properties, degree of crystallinity, oxidation state, and molecular weight distribution.

In nature, biotic and abiotic factors can act together to decompose organic matter. This is because some microbes excrete extracellular enzymes that act directly on plastics, and prior fragmentation and reduction of molar mass are not necessary to make the microorganisms available. An example of this is the degradation of polyhydroxybutyrate (PHB) by the action of intracellular and extracellular depolymerase of bacteria and fungi [18]. However, abiotic factors weaken the polymer structure, producing smaller polymer fragments that can pass through cell membranes and are biodegraded within microbial cells by cellular enzymes, catalyzing the biological stage of biodegradation. Most plastics degrade first at the polymer surface, as it is the most exposed and vulnerable to chemical (abiotic) or bacterial/enzyme (biotic) attack. The Table 1 presents a list of enzymes and bacteria involved in the biodegradation of various types of polymers, including the type of polymer, biodegradation mechanism, mode of action and mechanisms.

**Table 1.** Enzymes and bacteria involved in biodegradation of polymers.

| Type of Enzyme/Bacteria | Polymer Type | Biodegradation Mechanism | Mode of Action and Mechanisms |
|---|---|---|---|
| Proteases | Proteins | Hydrolysis | Catalyze the cleavage of peptide bonds in proteins, breaking them down into smaller peptides and eventually amino acids. |
| Lipases | Lipids | Hydrolysis | Break down ester bonds in lipids, producing free fatty acids and glycerol. |
| Amylases | Starch | Hydrolysis | Break down the $\alpha$-1,4-glycosidic bonds in starch, producing glucose. |
| Cellulases | Cellulose | Hydrolysis | Break down the $\beta$-1,4-glycosidic bonds in cellulose, producing glucose. |
| Chitinases | Chitin | Hydrolysis | Break down the $\beta$-1,4-glycosidic bonds in chitin, producing N-acetylglucosamine. |
| Laccases | Lignin | Oxidation | Oxidize the phenolic and non-phenolic structures in lignin, breaking down the polymer into smaller fragments. |
| Peroxidases | Lignin | Oxidation | Catalyze the oxidation of lignin by hydrogen peroxide or oxygen, breaking it down into smaller fragments. |
| *Kosakonia* sp. | Polyethylene | Anaerobic metabolism | Production of extracellular enzymes to break down polyethylene into smaller fragments for cellular uptake and utilization as carbon and energy sources. |
| *Aspergillus* sp. | Various | Aerobic metabolism | Produce reactive oxygen species and a range of extracellular enzymes, e.g., cellulases, hemicellulases, and ligninases. |

The data in the table were constructed based on references [18–20].

On the other hand, biotic degradation is classified as the biodegradation caused by the action of microorganisms that modify and consume the polymer or polymeric monomers, producing molecules of low molar mass (acids, aldehydes, terpenes, and $H_2O$) and gases ($CO_2$, $CH_4$, and $N_2$). According to Oliveira et al. (2020), the main biodegradation mechanism is the adhesion of microorganisms to the polymer surface, followed by the colonization of the exposed surface. After colonization, enzymatic degradation of the

polymer occurs by hydrolytic cleavage, producing molecules of low molecular weight until the final mineralization in $CO_2$ and $H_2O$ [21,22].

### 2.3. Stages of Biodegradation

Biodegradation can occur over different periods (as long as it meets the established standards, typically around 6 months) in various circumstances and environments. Ideally, it should happen naturally, without human intervention. The stages of biodegradation of polymeric materials are categorized into four stages: (bio)deterioration, (bio)fragmentation, assimilation, and mineralization (Figure 3c). The process can stop at any stage; however, plastic biodegradation is only confirmed after verifying mineralization [22,23].

### 2.3.1. (Bio)Deterioration

The first indication of biodegradation is (bio)deterioration, in which the cooperative action of different microorganisms and/or abiotic factors fragments macro materials into small fractions (micro, sub-micro). Deterioration is a superficial degradation that can be identified with the naked eye and is responsible for modifying the material's mechanical, physical, and chemical properties. The big difference between biodegradation and deterioration is that the former is only confirmed by deterioration, while the latter is already observed by weight loss and macro-deformations (cracks, roughness, scratches, holes).

### 2.3.2. (Bio)Fragmentation

The second stage is biofragmentation, a step in which catalytic agents (e.g., enzymes) are excreted by the microorganisms, progressively reducing the molecular weight of the polymers. At this moment, the polymers are cleaved until the production of small molecules (dimers and monomers). The term biofragmentation or, in some cases the depolymerization, should be used for situations where macromolecular size reduction occurs without changing the chemical composition or the monomer unit's structure. Some enzymatic tests can be used to estimate the propensity for biofragmentation of polymers, such as tests of enzymatic mixtures for solid-wet reaction in polyethylene terephthalate (PET) [24].

### 2.3.3. Assimilation

Assimilation is the third stage and occurs in the cytoplasm when small molecules produced in depolymerization integrate with the microbial metabolism to produce energy, biomass, and other metabolites. Therefore, assimilation happens when microorganisms use polymers as their carbon/nitrogen sources, converting $CO_2$ or $CH_4/NH_3$ or nitrate into cell building blocks [10,25]. This assimilation can occur through the three classic catabolic pathways: aerobic respiration, anaerobic respiration, and/or fermentation, and it is the only event in which fragments of polymeric materials are absorbed inside microbial cells [22]. This absorption is responsible for producing energy, via the production of adenosine triphosphate (ATP), aiming to form structural elements of cells. This allows microorganisms to grow, proliferate, and consume new energy packages (substrates) from the environment [22].

### 2.3.4. Mineralization

The final stage, mineralization, occurs concurrently with assimilation, during which organic material is converted into minerals through the excretion of metabolites and simple molecules that can be absorbed by both the environment and microorganisms [18,22]. The biodegradation process typically involves different microorganisms with complex interactions and symbiosis, making it difficult to simulate degradation in a natural environment in the laboratory. For instance, some microorganisms mainly break down polymers and produce $CO_2$ (mineralization), while others reduce the polymer into its constituent monomers, and some use these monomers and excrete simpler residual compounds that serve as substrates, while others use the excreted residues as a source of energy. With the metabolic routes' complexity and generation of new products, it is noteworthy that

$CO_2$ and $H_2O$ gases are produced during aerobic biodegradation, which can be used to monitor activity at this stage. In contrast, to aerobic processes, which produce $CO_2$, the anaerobic process results in the generation of both $CO_2$ and $CH_4$ [25,26]. Therefore, mineralization is the only stage capable of indicating the material's biodegradation and must be estimated through standardized respirometric methods, such as measuring the evolution of the gases mentioned above for anaerobic environments or oxygen consumption for aerobic environments, as in the ISO 14852.

*2.4. Greenwashing Concept*

Despite efforts to assess biodegradation and reduce environmental damage related to the improper disposal of plastic artifacts, in recent years, there has been an increase in the number of corporations that adopt green marketing strategies and label products with more environmental benefits than they actually have, e.g., biodegradability and sustainability. Greenwashing consists of deceiving consumers regarding the environmental conduct of a company or the environmental benefits that a product or service can offer. In this context, it is common to find packages with green parts, drawings of leaves, and words such as "green", "bio", and "eco", without explaining exactly what they refer to, but conveying the idea of a product that is less harmful to the environment. Such practices have been used to attract consumers who are aware and committed to sustainable actions, unduly influencing their purchasing decisions. Therefore, in addition to adopting norms for evaluating the biodegradation of plastics, inspections of commercialized products must be implemented to combat this practice.

As a result, some technical standards of biodegradability were developed to regulate the correct labeling of these materials and serve as references for assessing the level of degradation of plastics. These standards use a set of instruments and techniques to simulate biodegradation conditions in the laboratory as closely as possible to real environmental conditions, indicating the level of degradation for each stage of biodegradation. An example of this is the weatherometer (accelerated aging test), an instrument that subjects samples to different temperatures, UV radiation, and humidity for a specific time, making it possible to expedite the comparison between the properties of plastics. In this sense, the main standards that track the biodegradation of plastics, along with the techniques and interpretations employed, will be presented below.

## 3. Standardized Norms to Evaluate Biodegradation

The biodegradation standards for plastic materials comprise two crucial categories: a biodegradation testing method and biodegradation performance indicators. The first category provides a technique for estimating biodegradation and establishes a test procedure that precisely duplicates the environment intended for deterioration. The second category determines biodegradation by establishing calculations or analyses. Both circumstances are necessary to assess the biodegradability of plastic materials [27].

Table 2 classifies the methodologies currently available for determining the biodegradability of polymeric materials according to their intended disposal location, i.e., soil, compost, or aquatic systems.

**Table 2.** Summary of ASTM and ISO standards for plastic degradation in soil, compost, and aquatic systems.

| Environment | Standard or Test Method | Analysis Time (Months) | Parameters Monitored | Interpretation of Results and Validity Criteria |
|---|---|---|---|---|
| Soil | ASTM D5988-18 | 6 | $CO_2$ evolution | The reference material should have undergone 70% biodegradation, and the amount of $CO_2$ released from the control reactors should be within 20% of the average. |
| | ISO 17556:2019 | 6 or until 24 | BOD; $CO_2$ evolution | The reference material should biodegrade above 60%, and the amount of $CO_2$ produced should be within 20% of the average. |

**Table 2.** *Cont.*

| Environment | Standard or Test Method | Analysis Time (Months) | Parameters Monitored | Interpretation of Results and Validity Criteria |
|---|---|---|---|---|
| Landfilling | ASTM D5526-94D | Until no significant gas production | $CH_4$ and $CO_2$ evolution | The test method measures the percentage conversion of organic carbon in the sample to carbon in gaseous form, with a minimum test duration of 7 days. The level of biodegradation is compared to a cellulose-positive control when it reaches 70% biodegradation. |
| Compost | ASTM D6400-21 | 3–6 | $CO_2$ evolution | After 180 days, at least 90% of the sample's organic carbon (either absolute or relative) should have transformed into $CO_2$. |
| | ASTM D5338:15 | 4 | Cumulative $CO_2$ production, DMR, CMR, GMR | The sample should produce less than 2 g of volatile fatty acids per kilogram of dry matter, achieve 70% biodegradation according to the reference material, and the deviation of the biodegradation percentage from the positive reference should be less than 20%. |
| | ISO 14855-2012 | 6 | $CO_2$ evolution | According to the reference material, the sample must biodegrade at least 70% after 45 days. The difference between the percent biodegradation of the reference material in different vessels must be less than 20% at the end of the test, and the blank inoculum should produce between 50 mg and 150 mg of carbon dioxide per gram of volatile solids after 10 days of incubation. |
| | ISO 17088:2021 | 6 | $CO_2$ evolution | After 180 days, at least 90% of the sample's organic carbon (either absolute or relative) should have transformed into $CO_2$. |
| | ISO 14855-2:2018 | 6 | $CO_2$ evolution | After 45 days, the reference material must exhibit biodegradation above 70%. |
| Aquatic systems | ISO 18830:2016 | 24 | BOD; static test conditions | The reference material must exhibit biodegradation above 60% after 180 days. The difference between the percentage of biodegradation of the reference material in different vessels should be less than 20% of the mean at the end of the test. |
| | ISO 19679:2020 | 24 | $CO_2$ evolution; static test conditions | The reference material must exhibit biodegradation above 60% after 180 days. The $CO_2$ released from the blank at the end of the test should not exceed 3.5 mg $CO_2$/g wet sediment after 6 months. |
| | ASTM D6691-17 | 3 | $CO_2$ evolution; static test conditions | The reference material must exhibit biodegradation above 70%. |
| | ASTM D7991-22 | 24 | $CO_2$ evolution; static test conditions | The reference material must exhibit biodegradation above 60% after 180 days. |
| | ISO 14853:2016 | 3 | $CH_4$ and $CO_2$ evolution, DIC; static test conditions | The determination of the ultimate anaerobic biodegradability of plastics by anaerobic microorganisms requires degradation greater than 70% of the reference material, while the pH of the medium must remain between 6 and 8. |

CMR: Cumulative measurement respirometric system; DIC: Dissolved inorganic carbon; DMR: Direct measurement respirometric system; GMR: Gravimetric measurement respirometric system.

### 3.1. Biodegradation Test in Soil and Landfilling

Along with the structural characteristics, several "environmental factors" also affect the rate of deterioration at any particular site, including temperature, humidity, pH, sunlight, oxygen, nutrients, and microorganisms [28].

One of the environments on earth with the greatest biological diversity is soil. It is believed that one gram of soil contains up to 1 billion bacterial cells, tens of thousands of different species, up to 200 m of fungal hyphae, and a variety of other creatures, such as nematodes, earthworms, and arthropods [29]. Thus, the degradation of macromolecular chains by the action of microorganisms in the soil is a complex process.

It is evident that the presence of different types of microorganisms and pH, as well as different soil samples, can cause the same polymer to exhibit distinct biodegradation behavior. According to Agarwal [30], to avoid confusion, it would be efficient to limit the certification to the specific application and soil type.

Different tests for determining the biodegradability of polymers in soil are explained in the literature, ranging from simple tests such as mass loss determination or biological $CO_2$ concentration tests. Therefore, there is a contradiction in which methodology laboratory research should be used.

For example, the initiation of degradation of polyvinyl alcohol (PVA) in soil-buried conditions can take 120 days, according to the study presented by Mittal et al. (2021) [31]. However, in another study, the PVA film achieved 15% mass loss in 12 weeks using a similar burial method [32]. In light of this, one must consider that each polymer is degraded in a specific manner, accounting for variable intrinsic factors such as crystallinity, chemical structure, molecular weight, surface area, crosslinks, and physical form, as well as being constrained by certain soil environmental factors like temperature, humidity, pH, oxygen availability, location, sunlight, and others [12,28].

The two most up-to-date standards for measuring the aerobic decomposition of plastic materials in soil are ASTM D5988-18 "Standard Test Method for Determining Aerobic Biodegradation of Plastic Materials in Soil" [33] and ISO 17556:2019 "Plastics—Determination of the Ultimate Aerobic Biodegradability of Plastic Materials in Soil by Measuring the Oxygen Demand in a Respirometer or the Amount of Carbon Dioxide Evolved" [34].

In summary, ASTM D5988-18 measures the degree and rate of aerobic biodegradability of plastic material in the environment relative to reference material by measuring the $CO_2$ produced by microorganisms when demineralizing the polymer. Similarly, ISO 17556-2019 specifies a technique for determining the ultimate aerobic biodegradability of plastic materials in soil by calculating the oxygen demand in a closed respirometer or the amount of carbon dioxide evolved.

The experimental setup is similar in both methods. Samples (cut to specific sizes) are mixed into the soil, which is usually sieved to remove particles larger than 2 mm. The mixture is then fed into flasks (incubators) and kept at temperatures ranging from 20 to 28 ± 2 °C, with a moisture holding capacity of 40 to 100%. The amount of carbon dioxide that evolved during the biodegradation process is monitored for six months and can be extended up to 24 months with the help of $CO_2$ meters. Subsequently, the $CO_2$ concentration is calculated based on a "blank" [35,36].

The current criteria should be used as a starting point for developing potential strategies and establishing a general direction. However, it is still necessary to specify the methods based on the type of material being analyzed and provide specific definitions for each class of polymers, while also emphasizing the complete biodegradation time. As Agarwal [30] argues, "the use of the general term 'biodegradable polymers' can be misleading when applied to the context of using that polymer in different environments".

### 3.2. Biodegradation Test in Compost

Composting is a natural process that occurs in either anaerobic or aerobic settings. In this process, various microorganisms such as bacteria and fungi convert organic matter into less complex elements such as humus, water, carbon dioxide, heat, or methane in the case of anaerobic circumstances. Composting is an immensely helpful method in the disposal issue as it produces a valuable organic amendment and substrate that can be reintroduced into the economic system, which also helps to reduce greenhouse gas emissions [37].

According to Table 2, the most commonly used standard methods are ISO 14855-1:2012, ISO 14855-2:2018, ISO 17088:2021, ASTM D6400:2021, and ASTM D5338-15 [38–40]. These methods simulate typical aerobic composting processes in soil with waste found in most municipalities and are, therefore, representative. The method used in these tests is designed to give a percentage indicative of the conversion of carbon to carbon dioxide when the study compound is degraded.

ISO 14855-1:2012 and ASTM D5338-15 quantify the percentage of $CO_2$ by titration with HCl using phenolphthalein, or as an option, gas chromatography is used. Part 2 of ISO 14855-2:2018 measures the mineralization of a sample by a gravimetric method.

The methods cited above are limited because there are disparities in conditions between home and industrial composting, which can lead to a significant difference in the character of polymer degradation. In addition, actual composting times at commercial composting facilities are much shorter than what is allowed under the protocol.

Another criticism is the temperature that the methods require. Considering domestic conditions, i.e., temperatures below 58 °C, there will be a difference in the biodegradation of polymers. Investigating the degradation of PLA, Rudnik and Briassoulis [41] concluded that PLA is a compostable material in industrial composting facilities, but it will not disintegrate fast enough in home composting, as the minimum required conditions are usually not met.

Other methods determine the degree of disintegration of plastic materials, such as ISO 16929:2021. This approach is more practical and could be a good substitute because it is straightforward and affordable. However, this method is not meant to assess the biodegradability of plastic materials in composting environments. Further testing will be required before compostability can be claimed [42].

For broader objectives, two additional standards have been created: ASTM D6400-21, "Standard Specification for Labeling of Plastics Designed to Be Aerobically Composted in Municipal or Industrial Facilities" [43], and ISO 17088:2021, "Plastics—Organic Recycling—Specifications for Compostable Plastics" [44]. The ASTM D6400-21 is a biodegradability test that includes elemental analysis, plant germination (phytotoxicity), and mesh filtration of the resulting particles. In other words, this standard technique has more goals and analyses than ASTM D5338-15 [40]. ISO 17088:2021 covers adverse impacts on the composting process, facility, and compost quality, including excessive quantities of controlled metals and other dangerous components. As stated by ASTM D6400:2021 and ISO 17088:2021, the sample is classified as positive (compostable in industrial plants) if it satisfies the following standards taken together: (1) Within 12 weeks, there should be no more than 10% of the pieces larger than 2 mm remaining, and 2) 90% of the organic carbon should be transformed into $CO_2$ within 180 days (24 weeks).

The guidelines for assessing biodegradability in soil or compost make it evident that the methods could be clearer and more practical. Conducting a complete analysis requires a variety of expensive equipment and a laboratory environment that conforms to all regulatory criteria. Additionally, it is necessary to specify precisely the soil or compost used, as well as its organic matter content, N, P, and K content, among other elements. Therefore, regulations should evaluate biodegradability more broadly, on several levels and under a variety of environmental situations [30,45].

### 3.3. Biodegradation Test in Aquatic Systems

Studies have shown that approximately 10% of plastic waste is dumped into the ocean, and the annual amount of plastic waste entering aquatic environments may rise to an estimated 23 to 37 million tons per year by 2040, up from 9 to 14 million tons per year in 2016, in the absence of critical actions or under a business-as-usual scenario [46,47].

The marine environment is composed of various habitats with diverse environmental conditions, making it much more complicated to estimate biodegradability in aquatic systems than in terrestrial systems [48]. Although no standards have been established for freshwater systems, ASTM and ISO standards for marine habitats have been proposed. ASTM D6691-17 is a "standard test method for determining aerobic biodegradation of plastic materials in the marine environment by a defined microbial consortium or natural seawater inoculum" [49]. This test involves selecting and characterizing carbon content and molecular weight from plastic materials by preparing a uniform inoculum of various isolated marine microorganisms. Using a respirometer, it is possible to measure the total biogas ($CO_2$) produced as a function of time to assess the degree of biodegradability.

In addition, ASTM D7991-22 replicates the natural habitat present in the tidal zone [50]. This test is used to assess the extent of plastic biodegradation after being exposed to sand-filled silt that has been kept moist with seawater. The degree of biodegradation is calculated in the same way as the standard described above (ASTM D6691-17).

ASTM D7473/D7473M-21 is another standard test method that evaluates the weight attrition of non-floating plastic materials by open system aquarium incubations [51]. This method determines how much weight is lost over time when non-floating plastic materials (including formulation additives) are incubated under changing, open marine aquarium conditions. The purpose of this test is to gather the information that may be used to evaluate the possibility of the test material physically degrading.

Two more comprehensive standards, ISO 19679:2020 and ISO 18830:2016, are used to investigate aerobic biodegradation of non-floating plastic materials in a seawater/sediment interface [52,53]. The former measures the degree of biodegradation when settled on sandy marine sediment at the interface between seawater and the seafloor by measuring the evolved carbon dioxide ($CO_2$), while the latter measures biodegradation by the oxygen demand in a closed respirometer in the same environment.

The ISO 22766:2020 "Plastics—Determination of the degree of disintegration of plastic materials in marine habitats under real field conditions" is another standard that shows ways to measure the degree of degradation of plastic material in real field environments exposed to marine ecosystems but is not a biodegradation test [54].

It is difficult to predict how plastics will degrade in the marine environment, much like determining biodegradability in soil or compost. Biodegradability varies depending on the substance and surrounding environment of the system of interest. To account for any variations, both in the laboratory and in the field, different standardized tests can be evaluated concurrently to reproduce as much as possible the complex events of biodegradation and disintegration that occur in the environment, bringing the reality of the ecosystem as close as possible within the laboratory.

## 4. Methods and Analytical Tools for Evaluating Polymers' Biodegradation Process

As previously mentioned, there are various protocols for investigating the biodegradation process in polymers. Tests conducted in different conditions, such as soil, compost, aquatic systems, and accelerated degradation in a climate chamber, as well as at different temperatures, levels of UV exposure, and pH, are usually implemented to evaluate the stage and rate of biodegradation in the material of interest. Overall, there are some evidence by which plastics degradation can be observed in order to follow the biodegradation process.

- Loss of mass over time.
- Alterations in surface morphology, e.g., increased roughness, formation of pits or cracks.
- Changes in surface energy or wettability.
- Modifications in color or appearance.
- Changes in mechanical properties, e.g., decreased tensile strength or elongation at break.
- Changes in thermal properties, e.g., Tg or melting point (Tm).
- Alterations in chemical structure, e.g., molecular weight or functional groups.
- Release of degradation products, e.g., monomers or oligomers.
- Release of gases, e.g., carbon dioxide or methane.

With this in mind, several techniques can be employed together to better estimate the biodegradation process, understand the reactions, interactions, and changes that occur in the material structure during the different stages of the phenomenon, and predict the material's environmental impact. In Table 3, we summarize some recent studies regarding polymer biodegradation and the set of methods used.

**Table 3.** Methods used to investigate the occurrence of the biodegradation process in different polymers when exposed to varying conditions.

| Polymer | Degradation Test Condition | Tests Used to Assess Biodegradation | Reference |
|---|---|---|---|
| Poly(butylene adipate-co-terephthalate) (PBAT) | Soil (6 weeks at 25 °C) | SEM and isotope-specific quantification of $^{13}CO_2$ through NanoSIMS | [55] |
| Cellulose acetate (DS 2.5) | Several aqueous environments (12 months) | SEM, mass loss, FTIR, UV-Vis spectroscopy, TG, DSC, XRD, NMR, SEC | [28] |
| Polyhydroxyalkanoate (PHA), polybutylene succinate (PBS), polybutylene adipate-terephthalate and polylactic acid blend (PBAT/PLA), and polyester | Soil (25 °C and 37 °C up to 270 days) | SEM, $CO_2$ evolution by titration and gas analyzer | [33,38,39] |
| Poly(lactic acid) and chitosan composite | Active soil and sterile soil (25 °C up to 200 days) | Mass loss, tensile testing, molecular weight by GPC, FTIR, SEM, DSC, contact angle | [56] |
| Thermoplastic starch-graphene composites | Soil (23 °C, 120 days) and aerobic composting process | SEM, mass loss, $CO_2$ evolution | [57] |
| Polyethylene (PE), compostable bags (at least 60% of starch), and cellulosic plates | Anaerobic conditions | Biochemical methane potential and visual observation | [58] |
| PE and PE-modified with oxo-biodegradable compound | Accelerated weathering (UV irradiation 8 h/70 °C followed by a steam condensation 4 h/55 °C), and soil burial (30 °C) | Tensile testing, mass loss, contact angle, FTIR | [59] |
| PLA-clay composites | Aerobic composting | Mass loss, cumulative $CO_2$, visual observation | [60] |
| Low-density polyethylene (LDPE) | Solid mineral salt medium (Petri dish) | Tensile testing, SEM, mass loss | [61] |
| Polyvinyl alcohol (PVOH) incorporated with cellulose nanocrystals | Soil burial (3 months) | Tensile testing, FTIR, mass loss, DSC, SEM | [32] |
| Poly-d-lactic acid (PDLA) and cellulose microfibers | Solid and liquid mineral salt medium (Petri dish) | Mass loss, visual observation, FTIR, SEM, NMR | [62] |
| Starch and gelatin bioplastics | Soil respiration chambers | Respirometry (OxiTop) | [63] |

SEC: scanning electron microscopy; NanoSIMS: nanoscale secondary ion mass spectrometry; FTIR: Fourier transform infrared spectroscopy; TG: thermogravimetry; DSC: differential scanning calorimetry; XRD: X-ray diffraction spectroscopy; NMR: nuclear magnetic resonance spectroscopy; SEC: size exclusion chromatography; GPC: gel permeation chromatography.

### 4.1. Macro and Microscopic Changes

Visual examination of the degradation process enables macroscopic changes in the material's color, transparency, aspect, and size to be observed. To illustrate this point, Figure 4a,c,e,g depict the changes that took place in zein and cellulose acetate films after burial in soil. Color and transparency changes may occur due to the oxidation of specific functional groups, loss of additives, presence of external particles adhered to the polymer's structure, and formation or bleaching of chromophore molecules [64,65]. The naked eye can detect the phenomenon, and qualitative observations can often be reported to readers. For example, Andersson et al. (2010) [65] observed that plasticized poly(L-lactide) films, initially transparent, became whiter after they were exposed to hydrolytic degradation, and the authors reported their visual observations.

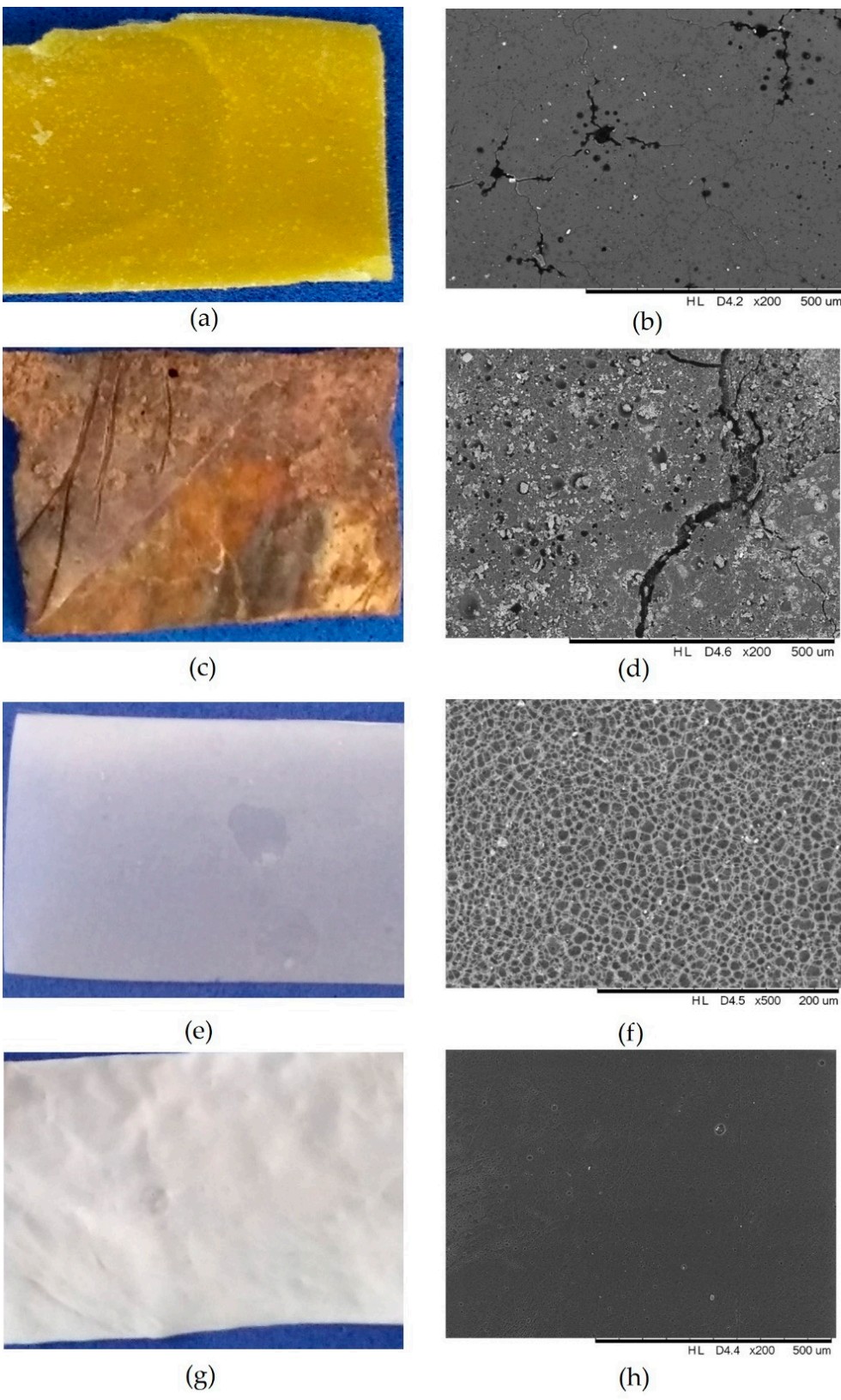

**Figure 4.** Photographs (**a**,**c**,**e**,**g**) and SEM images (**b**,**d**,**f**,**h**) of polymeric films: zein film plasticized with tributyrin before (**a**,**b**) and after burial in soil for 60 days (**c**,**d**); cellulose acetate film plasticized with glycerol before (**e**,**f**) and after burial in soil for 60 days (**g**,**h**). Source: elaborated by the authors.

The use of non-destructive analytical techniques can contribute to a better evaluation of the results, as they provide a more reliable measure of the observed color/aspect change. In this regard, the yellowness index and lightness value (b* and L* in CIELAB, respectively) are often used to assess discoloration in polymers. Pastorelli et al. (2014) [66] evaluated the occurrence of color modification in different polymeric materials (such as cellulose acetate propionate, polycarbonate, and polyurethane) that were exposed to several conditions (humidity, heat, light, and pollution) for over a year. The authors noticed that, aside from a few samples, the investigated polymers exhibited extensive discoloration, confirmed by the high Δb* and ΔL* values obtained. Similarly, Ammala et al. (2002) [64] subjected polypropylene and high-density polyethylene samples to accelerated weathering and used both visual observations and the yellowness index to better assess the color modifications in the polymers due to UV degradation.

Microscopy techniques can be valuable allies when assessing the degradation of polymers. For example, scanning electron microscopy (SEM) and atomic force microscopy (AFM) allow for the visualization of localized degradation on the polymer surface at the micro or nanoscale. Important details, such as erosion, cracks, fractures, wrinkles, holes, and even microbial colonization and loss of additives that may go unnoticed by the naked eye, can be evaluated with the aid of these tools [67–70]. For instance, SEM images displayed in Figure 4b,d,f,h allow the observation of fractures and an increase in the number of holes in the zein film after burial in soil, as well as the loss of plasticizer in the cellulose acetate film under the same condition. Vasile et al. (2018) [56] observed that after 150 days in soil, the surface of poly(lactic acid) films changed from smooth to rough, with several cracks and holes appearing. Kaczmarek-Szczepanska et al. (2021) [71] noticed bacterial biofilm on the surface of chitosan/tannic acid materials under degradation in soil and compost after 14 days. Damages due to microbial adhesion and metabolism were also verified on the investigated samples.

Another macroscopic observation that is often verified when investigating bio-based polymers is the change in size and fragmentation throughout the biodegradation process [56,58,72–74]. This change is frequently reported using mass loss as a function of time, a simple technique widely used to assess the evolution of material degradation. Rogovina et al. (2013) [73] evaluated the biodegradability of different low-density polyethylene blends with cellulose, chitin, and chitosan in the soil through mass loss. The samples were weighed from time to time for 200 days, and the authors observed that ternary blends of cellulose, chitin, and polyethylene exhibited a more significant mass loss than the other blends. In turn, Bilo et al. (2018) [72] produced a new bioplastic from rice straw and studied its biodegradability in soil. Several macroscopic changes were verified over the 105 days of analysis, such as the appearance of incrustations and fragmentation. The authors used the results from the mass loss test to conclude on the material's decomposability feature.

However, it is important to bear in mind that the material can absorb moisture/water during the course of the analysis, especially if the polymer has a more hydrophilic nature, such as starch and cellulose. In this sense, an increase in mass will be observed [56,72]. Incrustation of particles and formation of biofilm may also increase mass. Therefore, it is essential to always correlate the visual changes in samples, such as the presence of microbial growth and swelling, with the mass loss results to avoid inconsistencies.

Drying samples before weighing them is another strategy used to minimize errors due to moisture absorption [74–76]. Another issue that may limit the method's accuracy is the fragmentation of samples. As stated by Quintana et al. (2013) [77], a high degree of material degradation may make it difficult to recover its fragments, which hinders the correct determination of mass. It is important to take this into consideration when interpreting the results of mass loss tests. Overall, it is crucial to combine different techniques to obtain a complete evaluation of the material's biodegradability and to consider the limitations of each method to avoid misleading conclusions.

### 4.2. Gas Evolution Methods

The aerobic biodegradation of polymers occurs through the action of microorganisms, which break down the organic components into biomass and $CO_2$. To measure the biodegradation rate, respirometric methods can be used, which focus on the increase in $CO_2$ or the consumption/demand of $O_2$ by the microorganisms involved. This method requires the presence of microorganisms, which can be naturally occurring or previously inoculated, as well as a chosen set of parameters such as the type of medium (soil, compost, aquatic system), temperature, pH, and humidity [55,71].

$CO_2$ detectors can be used to measure the respiratory activity of microorganisms in the system where the degradation occurs, and the results can be expressed as the accumulation of $CO_2$ over time [78]. The cumulative amount of $CO_2$ produced during the process can also be used to calculate the percentage of biodegradation [60]. Gas chromatography (GC) is another essential tool to detect and quantify the $CO_2$ produced by microorganisms during biodegradation, as reported by Rose et al. [79] (2020) and Kalita et al. (2021) [80]. Additionally, $CO_2$ concentration can be measured by titration with NaOH using methyl orange and phenolphthalein as indicators [81].

The biochemical oxygen demand (BOD) can be evaluated using specialized equipment such as the OxiTop®, which measures the pressure inside the vessel over time [71]. Since the system includes a $CO_2$ absorber, the pressure reduces over time due to $O_2$ consumption, and the equipment compares the obtained values with the initial ones, converting the difference into BOD [82]. Lastly, anaerobic degradation of polymers can occur due to the activity of anaerobic bacteria. In this particular case, methane accumulation is verified instead of $CO_2$ production or $O_2$ consumption. To estimate the biodegradation evolution, the biochemical methane potential (BMP) test can be used [58,83].

### 4.3. Methods Based on Mechanical Properties

Knowing the mechanical properties of polymers are essential to defining the material's application, measurements of resistance, hardness, stiffness, and tenacity are often performed to determine whether the polymer meets the specific requirements for proper use. Since these properties depend on the configuration of the polymer chains and the intermolecular forces that occur among them, as well as environmental parameters, such as temperature and humidity, it is possible to deduce that the degradation process will affect these attributes. For example, intimate and prolonged contact with water molecules, whether in aqueous media or soil moisture, can plasticize the polymer matrix, negatively affecting mechanical properties due to swelling and hydrolysis [84,85]. In this sense, these properties are commonly used to evaluate the progress of biodegradation in polymers.

Tensile strength is the most investigated parameter regarding mechanical properties when assessing the evolution of degradation in polymeric materials [56,61,78,86]. The assay can be performed on a tensile testing machine with an extensometer. The results will indicate the maximum load before fracture that the polymeric material can support when stretched. Sangale et al. (2019) [87] and Munir et al. (2018) [61] used the tensile strength values to determine the potential of some fungi to degrade synthetic polymers. For example, Sangale et al. (2019) [87] observed a reduction of almost 95% in the tensile strength of polyethylene when certain strains of Aspergillus were involved in the biodegradation process. Similarly, Munir et al. (2018) [61] found that the tensile strength of polyethylene decreased by around 40–60% when in contact with the tested fungi. Young's modulus and elongation at break are other mechanical parameters that can be evaluated. Vasile et al. (2018) [56] found that Young's modulus and elongation at the break of PLA blend films changed after the samples were buried in the soil. However, the authors mentioned that the changes depended on the composition of the film, especially in the presence of additives, e.g., plasticizers, fillers, reinforcements, and dyes. For example, the Young's modulus of pristine PLA films decreased over time, while the same parameter increased over time in plasticized and blended PLA films [56]. These results can be strengthened and better

understood when interpreted alongside findings obtained by other analyses, such as SEM, mass loss, thermal methods, and spectroscopic methods, which are covered below.

### 4.4. Methods Based on Thermal Properties

Thermal methods, such as thermogravimetry (TG), differential thermal analysis (DTA), and differential scanning calorimetry (DSC), are essential in determining material features concerning thermal properties. Concerning polymers' biodegradation, DSC provides valuable information about the materials' crystallinity. Some polymers typically present two domains: crystalline and amorphous. The amorphous phase is the first to be affected when the biodegradation process occurs due to the more disorganized state of the molecules than the crystalline portion. Thus, comparing thermal parameters' values related to the polymers' crystallinity (Tm, enthalpy of melting, and enthalpy of crystallization) before and after exposing the material to the biodegradation condition is an interesting method to assess the occurrence of the phenomenon. If the amorphous region is degraded, an increase in the polymer's crystallinity is expected [56,88,89].

TG is another valuable method for assessing biodegradation on polymers. Changes in the TG curve can indicate polymer structure modifications and additive loss [57,59,62,63,77,90]. High-resolution TGA allowed Quintana et al. (2013) [77] to quantify the amount of plasticizer lost when plasticized cellulose acetate films were subjected to accelerated weathering. Although it was not possible to determine if the plasticizer had degraded or just exudated from the film, the analysis can shed light on polymers' biodegradation, especially when it comes to composite materials, as it helps to identify whether the observed mass loss is due to polymer degradation or loss of additives.

### 4.5. Spectroscopic Techniques

Fourier transform infrared spectroscopy (FTIR), X-ray diffraction spectroscopy (XRD), and nuclear magnetic resonance spectroscopy (NMR) are indispensable techniques for assessing the biodegradation of polymers from a more chemical and structural point of view. For instance, FTIR can better elucidate the process on a molecular scale as the method can show chemical changes, such as the formation and disappearance of functional groups, which may indicate the occurrence of oxidation, moisture absorption, breakage of covalent bonds, and loss of additives [15,56,89,91]. In some cases, it can even confirm the involvement of microorganisms in the materials' degradation [90]. The technique can be combined with other methods; Liu et al. (2019) [91] used TG-FTIR to study plasticizer loss in cellulose acetate materials during aging. The combined techniques indicated that degradation happened and impacted the thermal stability of the polymer.

As stated in Section 4.4, DSC can be a valuable analysis for inferring the crystallinity of samples. Similarly, XRD is another method used to determine the crystallinity degree of the studied material, making it possible to calculate the percentage of amorphous and crystalline domains present in the sample and the crystallinity index [88,89]. Finally, NMR is a powerful spectroscopic method for studying the structure and composition of polymers. In biodegradation analysis, the technique can indicate changes in the material's structure, such as bond breakage of carbons in the side chains and backbone, and incorporation of oxygen molecules into the polymer [74,92,93]. When studying blend films composed of two or more polymers, NMR can indicate whether all polymers show the same biodegradation rate. For example, when studying blends composed of polystyrene and starch, NMR was crucial to confirm that only the starch component was degraded when the samples were buried in soil, while the polystyrene portion remained unaffected [92].

## 5. Prospects

The leakage of plastic into the environment is a significant issue related to the inappropriate disposal of end-of-life materials [94]. The traditional life cycle of most plastic materials is linear, with 79% of all plastic produced ending up in landfills or the environment, while the remainder is incinerated (12%) or recycled (9%) as part of a circular

economy approach [95]. During incineration, N-containing, S-containing, and Cl-containing polymers can produce toxic gases such as NOx, SOx, and HCl. Similarly, additives in polymers may release various toxic substances upon burning that require potentially expensive capture and treatment interventions [96].

The increasing concern regarding the environmental impact of plastic waste and the plastic emission of greenhouse gases is motivating the transition toward a "circular plastic economy". The circular economy is defined by the U.S. Environmental Protection Agency as a system "that involves industrial processes and economic activities that are restorative or regenerative by design," and it focuses on eliminating waste and optimizing materials production. The International Organization for Standardization (ISO) has developed a series of standards aimed at setting up an environmental management system to help implement a circular economy (ASTM, 2023) [97].

Recycling is often the first solution considered when contemplating a circular economy (ASTM, 2023). However, despite increased recycling efforts since the 1980s, non-fiber plastic recycling rates remain at only 18% [94]. The limited progress in the recycling process is mainly due to difficulties in identifying and separating different types of plastics based on chemical structure, contaminants, and molar mass. This is a significant barrier, particularly in the mechanical recycling process, which converts waste plastics into new shapes through mechanical force and heat. Although it is the simplest, cheapest, and most commonly used form of recycling, the quality of the resulting products is highly dependent on input quality and requires well-sorted and contamination-free plastic waste, which is often scarce [95,98].

In turn, chemical recycling depolymerizes the polymer to recover the monomers, which, after appropriate separation, can undergo polymerization into materials of defined quality [99]. However, the chemical process is complex and, thus, more expensive, particularly during the implementation phase, requiring financial incentives [95]. Additionally, chemical consumption can jeopardize the sustainability of the recycling processes.

In general, the recycling process faces the problems of difficult selective collection, reverse logistics needing to be well-established, and compromised labor conditions for the waste collectors [100]. During the separation process, even the plastics marked with the universal plastic resin symbol (numbers from 1–7) pose challenges for discrimination of the materials encountered. As an alternative, high technology such as near-infrared spectroscopy can selectively identify bioplastics through scanners; for example, polylactic acid can be identified with 98% accuracy [101]. Advanced sorting technologies include X-ray photoelectron spectroscopy (XPS) and laser-induced breakdown spectroscopy (LIBS), inert detectable markers in materials for "barcoding", and using artificial-intelligence-based robotic sorting [102], which are high-cost options too. However, those technologies are still far from the reality of the major industries.

A solution to increase recycling rates is to incorporate nanomaterials in monolayer films to achieve the performance of multilayer plastics that are difficult to separate and, therefore, not recyclable [95]. Colored or low-density materials (films, foams) and medical contaminants are further complications and can render products non-recyclable [103]. Food-grade recycled materials are, therefore, hard to obtain due to various organic and even microbiological contaminations [104]. Virgin polymers are often mixed with recycled materials to improve the quality of the recycled ones [98].

Alternatively, biodegradable plastics can be used as a solution for plastic materials to help achieve some of the world's sustainable purposes. Bioplastics that are 100% bio-based are currently produced at a scale of ~2 million tonnes per year and are considered a part of future circular economies [95]. Depending on the type, bioplastics can offer improved circularity by using renewable (non-fossil) resources, a lower carbon footprint, re-extrusion performance (laboratory stage), biodegradation as an alternative to end-of-life, and enhanced material properties [105]. However, these benefits are highly dependent on several factors, including the chemical structure, the manufacturing process, and the most likely end-of-life scenario. All these factors must be evaluated across the life cycle, along with metrics such as climate impact, ecotoxicity, and recyclability [100,106,107].

Like traditional plastics, bioplastics also raise concerns about the leaching of monomers, oligomers, and additives and, therefore, require the same scrutiny in product design and formulation [7]. However, given the trade-offs, implementing bioplastics faces several challenges [95]. Food packaging and fast-moving consumer goods are the largest markets for short-lived to medium-lived plastics and, therefore, also for bioplastics.

The problems associated with bioplastics are concentrated in their inferior properties, which can be solved by incorporating additives or reinforcing substances or by mixing with different polymeric resins, resulting in blends or composite materials with improved mechanical, thermal, and barrier properties [108]. Nevertheless, the diversity of the materials' composition makes the processing and standardization of the products in terms of quality and performance difficult due to the seasonality and environmental variations in which the precursors of the biopolymers undergo. From a technological point of view, most bioplastics require new equipment to manufacture, which means that a revolution should be made to scale up bioplastic production.

Biodegradation is a disposal option for easily hydrolyzable polymers, such as aliphatic esters like polylactic acid, cellulose materials, and starch films [109]. Biodegradation and composting describe the microbial digestion and metabolic conversion of polymeric material into $CO_2$, $H_2O$, and other inorganic compounds [110]. Despite earlier hopes, biodegradation is non-trivial, as the rate of biodegradation is highly dependent on a polymer's chemical structure, stabilizing additives, the surrounding conditions (such as the presence of $H_2O$ and $O_2$), and the microorganisms in the soil [111]. These conditions are often not met in home compost, open water, or industrial composting facilities. Composters often reject biodegradable plastics as required decomposition times exceed typical composting process times of 6–8 weeks [112].

Biodegradable polymers can compost anaerobically to $CH_4$, which has a greenhouse gas impact that is 20 times higher than that of $CO_2$ [113]. However, the $CH_4$ produced can be captured and burned to produce $CO_2$, $H_2O$, and heat for industrial purposes [114]. Anaerobic digestion is feasible for several polymers, including thermoplastic starch, polycaprolactones, polyhydroxyalkanoates, and polylactic acid at elevated temperatures [95].

The greenhouse gas impact is not the only hazard associated with plastics, including bioplastics and plant-based materials. Zimmermann et al. (2020) [7] evaluated the toxicity of different bioplastics, performed according to an ISO guideline (ISO 11348-3, 2017) [115], in which the bioluminescence inhibition of *Aliivibrio fischeri* was evaluated as an indicator for baseline toxicity [116]. Two-thirds (67%) of the 43 bioplastic extracts induced baseline toxicity. All cellulose-based and starch-based samples, as well as the polyhydroxyalkanoate samples, inhibited bioluminescence, mostly with a high potency and effect level. The authors also observed that the materials activated oxidative stress response in human cells. Their study demonstrated that bio-based or biodegradable materials available on the market, such as extract shape, are just as toxic as conventional plastics regarding the chemicals they contain. This highlights that the positive connotation of "biological" or "sustainable" materials does not extend to chemical hazards. The authors suggested future works such as migration studies with food simulations or in environmental conditions to identify the toxicity and chemicals migrating under real-world conditions to estimate human exposure to those.

The problems associated with the incomplete biodegradation and the formation of microplastics are becoming increasingly concerning, particularly with regard to their presence in water and soil ecosystems [117,118]. Recent research has focused on detecting, characterizing, and evaluating the toxicology of microplastics in marine and freshwater ecosystems. Microplastics are a heterogeneous mixture of plastics that are less than 5 mm in diameter, including plastic fibers, granules, and fragments, and are considered emerging contaminants of concern [119]. According to the latest global estimate, 93–236 thousand tons of microplastics are floating on the ocean surface, corresponding to as many as 51 trillion particles [120]. Growing evidence has also shown that microplastics are present in terrestrial ecosystems [121], and that 79% of global plastic waste is stacked in landfills [122].

Microplastics have been demonstrated to cause oxidative and pathological stress, reduce immune functions, and cause cancer in marine biota. Other studies have evaluated the effects of these small plastic particles on the feeding, predation, and reproductive activities of pelagic and benthic species [117].

Biodegradation of polyolefin materials is even more challenging because they lack cleavable functional groups along their backbones, are highly hydrophobic, have a high molecular weight, and contain stabilizing additives [112]. Small fragments less than 5000 Da are believed to be metabolized by some organisms; however, the molecular weight of most polyolefin plastics is millions of Daltons. Partial biodegradation (5–20%) of polyethylene films by waxworm bacteria and Pseudomonas strains has been observed, occurring over 1–2 months [123]. Non-degradable polymers, such as polyethylene furanoate, can be made more degradable by copolymerization with more hydrolyzable, hydrophilic, and less crystalline copolymers [124]. However, copolymerization can negatively affect the material's properties. Traditional polymers, such as polyolefins, can also be blended with biodegradable polymers, such as starch, protein, or natural fiber, to increase the material's susceptibility to biodegradation. Nevertheless, it remains unclear whether such compounds decompose into sufficiently small particles or merely fragment to form microplastics. There is a brand of organic additives, Eco-One®, which promises to enhance the biodegradation of traditional plastic products in a biologically active landfill for 1–12 months, broadening the options in the market [125].

The biodegradation process can be aided by physical processes, especially those that help with the fragmentation and reducing particle size [95,126]. For example, amorphization of crystalline structures in typically semi-crystalline plastics through micronization or extrusion can make them more susceptible to enzymatic degradation [125]. Hydrolysis cleaves susceptible bonds in accessible amorphous regions of a polymer; typically, aliphatic esters, microbial enzymes, and acids or bases can enhance hydrolysis. Photodegradation using UV light breaks tertiary and aromatic C–C bonds. This process can be enhanced by embedding metallic catalysts in the polymer [127]. Similarly, oxo-degradation (that is, decomposition by oxidation) can be triggered by metals; however, this can lead to fragmentation into microplastics and insufficient digestion [128].

Therefore, biodegradable materials and their safe and eco-friendly properties are difficult to certify and should be carefully assessed. To this end, the American Society for Testing and Materials (ASTM), a developer of international voluntary consensus standards, regularly delivers standards, test methods, specifications, guides, and practices to standardize and guide plastic tests. Several methodologies have been recommended by the ASTM to standardize the test methods to verify the sustainability of plastic materials, such as the ASTM D5338-21 [129] standard biodegradation test, which measures aerobic biodegradation of plastic materials under controlled composting conditions, the ASTM D6400-21 compostability test method [43], the ASTM G160-12 (2019) [130] standard practice for evaluating microbial susceptibility of nonmetallic materials by laboratory soil burial, the ASTM D5526-18 standard methods for determining anaerobic biodegradation of plastic materials under accelerated landfill conditions, the ASTM D5988-18 standard test method for determining aerobic biodegradation of plastic materials in soils, the ASTM D5511 standard test method for determining anaerobic biodegradation of plastic materials under high-solids anaerobic digestion conditions, and others. However, these standards are general, and there are no well-defined parameters to consider plastic biodegradable or safe. Sometimes, it is difficult to understand and reproduce the same conditions of the test. In the name of sustainability, several major organizations and companies have committed to developing and producing more sustainable plastics, which are set to increase future bioplastic demand. However, investment and scaling of bioplastic technologies remain a high-risk business, with the central problem of uncertain demand due to high prices and undefined end-of-life treatment. Nevertheless, larger scales could reduce costs and create demand and incentives for recycling infrastructure. Moreover, the threat of rising oil prices

due to a supply shortage, once advertised as the main driver for renewable-resource-based materials, has not materialized [131].

From a political perspective, the World Economic Forum (WEF), the Ellen MacArthur Foundation, and McKinsey & Company are promoting science-based policy initiatives for a circular plastics economy. Recommendations include adopting extended producer responsibility (EPR) schemes and clearer labeling standards for bioplastic materials. In addition, the UN Industrial Development Organization, the Group of Twenty (G20) nations, and the Plastic Waste Partnership are collaborating on circular economy measures. The proposals include reducing waste exports into countries with high leakage rates by 90%, doubling global mechanical recycling capacity, improving design-for-recycling to expand global recyclable plastic from 21% to 54%, and implementing known solutions to eliminate major micro-plastic sources [95].

Despite all the technological concerns, the government should provide arrangements to boost the circular economy, such as raising environmental awareness, providing financial support, offering legislative incentives, and inspecting behavior. Educating consumers and companies about "life cycle thinking" will encourage a holistic view of plastic products beyond their obvious impacts associated with the use and disposal of plastics [95]. In this context, the sustainable harvesting and catalytic conversion of local, non-food, renewable resources, and biological wastes into bio-based plastics can provide greater sustainability than established fossil fuel extraction and refining practices [132]. Moreover, useful measures towards future circular economies include a drastic reduction in plastic consumption, designing products that can be reused and recycled in their markets, improving process energy efficiency in plastic and bioplastic manufacturing combined with the use of renewable power, increasing collection rates and market penetration of robust and circular recycling and "upcycling" methods [133].

In turn, the understanding of biodegradability of polymers among the general public is currently low due to several reasons, including lack of education and awareness, complex scientific terminology, and limited access to accurate information. To boost understanding, a disruptive way would be to use innovative communication strategies that simplify complex scientific concepts and make them easily accessible to the public. To address this issue and promote greater public awareness of polymer biodegradability, innovative communication strategies could be employed, such as those presented below.

- Public education campaigns: Develop a targeted public education campaign focused on promoting awareness of biodegradable polymers, their benefits, and their proper disposal. This could include media outreach, social media campaigns, and public events.
- Collaborations with industries: Partner with industries that use polymers to promote the use of biodegradable alternatives and educate consumers on their benefits.
- Improved labeling: Develop standardized labeling for biodegradable polymers that is easy for consumers to understand and includes information on proper disposal.
- School curriculum: Incorporate the science of biodegradable polymers into school curriculum, starting at a young age. This can help to create a more informed and environmentally conscious future generation.

This review emphasizes that, until now, none of the existing materials and processes established are sustainable. However, recent paradigm shifts, such as moving from conventional plastics to renewable or biodegradable plastics in single-use materials—e.g., disposable cups and plates—indicate that the world is on the right track to satisfy consumption with the survival of the world. Nevertheless, changes in people's lifestyles and government support are paramount.

## 6. Final Considerations

This review provides an overview of the main stages and mechanisms of polymer biodegradation. Despite the increasing interest and progress in biodegradable polymers, there are still several limitations that need to be addressed to promote their widespread

adoption as a sustainable alternative to conventional plastics. The following points outline important areas where further research is necessary to make progress in the field of biodegradable polymers.

(i) Investigation of the effects of additives and contaminants on biodegradation: The majority of plastics contain additives that can affect their properties, e.g., color, flexibility, and durability. However, these additives can also interfere with the biodegradation process, and in some cases, release toxic substances. Further research could investigate how different types of additives and contaminants affect biodegradability, and develop strategies to remove or mitigate their impact.

(ii) Use of microbial consortia and genetic engineering to enhance biodegradation: Microbial consortia are communities of microorganisms that work together to degrade complex compounds. Genetic engineering techniques can be used to modify microorganisms to enhance their biodegradation capabilities. Research in this area could involve identifying new microbial consortia and engineering them to enhance their ability to degrade specific types of polymers.

(iii) Advanced materials design: There is an opportunity to develop advanced materials with superior biodegradability and functionality. One approach is to design polymers with built-in degradation mechanisms, such as responsive polymers that break down in response to specific stimuli. Another approach is to develop materials that are composed of multiple types of polymers, each with different degradation rates, which could enable more precise control over the biodegradation process.

(iv) Development of circular economy strategies: In addition to developing new biodegradable polymers, research could focus on strategies for closing the loop on plastics. This could involve developing new recycling technologies that can efficiently recycle a wide range of plastics, as well as exploring alternative uses for waste plastics, such as energy recovery or conversion into other materials.

(v) Synthetic biology: Synthetic biology is a rapidly advancing field that involves the design and construction of new biological systems for specific applications. In the context of polymer biodegradation, synthetic biology could be used to engineer microorganisms with enhanced biodegradation capabilities, or to develop synthetic enzymes that can break down specific types of polymers.

(vi) Synthetic biology for upcycling: In addition to engineering microorganisms to enhance biodegradation capabilities, synthetic biology can also be used to engineer microorganisms to upcycle plastics into new materials with higher value. For example, using bacteria to convert plastics into renewable chemicals that can be used to make new plastics or other materials.

(vii) Nanotechnology: Nanotechnology involves the manipulation of materials at the nanoscale to create new properties and functionality. In the context of polymer biodegradation, nanotechnology could be used to develop new materials with enhanced biodegradability or to create new methods for breaking down plastics into their constituent parts. For example, developing nanocatalysts that can accelerate the breakdown of plastics and promote mineralization.

(viii) Waste-to-energy technologies: In the context of biodegradable polymers, waste-to-energy technologies could be used to convert biodegradable plastics or waste materials into energy, such as biogas or biofuel, through anaerobic digestion or other processes. This could provide a more sustainable and efficient way to dispose of biodegradable plastics than landfilling or incineration.

Furthermore, it is also important to note that to classify plastics as biodegradable, it is not enough to only observe a loss of material mass or partial disintegration, but rather the assimilation and mineralization of the plastic by microorganisms within a reasonable time frame (approximately 6 months). However, even when the four stages of biodegradability are validated, there is no guarantee that the polymer will biodegrade when mixed with other polymers after improper disposal or when contaminated by chemicals or food. Therefore, it is essential to establish centralized and rigorously revised protocols for joint evaluations

of biodegradability and toxicity, given the advances in technology. It is not acceptable for a "biodegradable" plastic to release over 90% of $CO_2$ while simultaneously releasing toxic substances (e.g., bisphenol A) into the environment. Despite this, as biodegradable plastics continue to progress and gain attention in the market, they have the potential to mitigate the damage caused by the global issue of plastic pollution. Recycling strategies, circular economy, and conscientious consumption should also be considered viable alternatives for promoting sustainability and combating plastic pollution. Ultimately, we must remember that there is no Planet B, and it is our responsibility to take action to protect and preserve our planet for future generations.

**Author Contributions:** Conceptualization, R.R.A.S. and C.S.M.; writing—original draft preparation, R.R.A.S., C.S.M., S.C.T., T.R.A. and T.V.d.O.; writing—review and editing, R.R.A.S., C.S.M. and T.V.d.O.; supervision, R.R.A.S., C.S.M. and T.V.d.O. All authors have read and agreed to the published version of the manuscript.

**Funding:** This research was funded by the Coordenação de Aperfeiçoamento de Pessoal de Nível Superior—Brasil (CAPES, grant number: 200337/2022-0). Also, this study was financed in part by the Coordenação de Aperfeiçoamento de Pessoal de Nível Superior—Brasil (CAPES)—Finance code 001.

**Institutional Review Board Statement:** Not applicable.

**Informed Consent Statement:** Not applicable.

**Data Availability Statement:** The data presented in this study are available on request from the corresponding author.

**Acknowledgments:** The authors are grateful to the colleagues from the Food Packaging Laboratory of the Federal University of Viçosa (UFV) for exchanging experiences and knowledge.

**Conflicts of Interest:** The authors declare no conflict of interest.

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
