# Peer review of "Biodegradation of Polymers: Stages, Measurement, Standards and Prospects"

_2673-6209, doi:10.3390/macromol3020023_

Round 1

Reviewer 1 Report

The manuscript “Biodegradation of Polymers: Stages, Measurement, Standards  and Prospects" is interesting for the scientific community. The subject is well within the aims and scope of Macromol. Text, objectives, experimental design, results and discussions are well designed, consistent and well described. References are consistent and updated.I strongly recommend this article is worthy to publish in the journal after correction of some points described bellow:

1. Figures 1, 6 and 7 should be removed from the manuscript. Figures do not add anything to the article, they only add to the volume of the manuscript, which is already very large.

2. Line 320. The sentence is wrong. Methanation is a chemical reaction that converts COx to methane. The process described by the authors is anaerobic digestion. This process produces methane, but also CO2. This part of the manuscript needs to be corrected.

3. The use of SEM to assess biodegradation efficiency is debatable and questionable. It is not an objective method. The operator can always find an image that confirms his research hypothesis. I propose to remove information on this subject from the work (including Fig.5, which does not prove anything), or only to indicate that such a method also exists. Moreover, the work is very extensive.

Author Response

Reviewer 1:

Thank you very much for your consideration and suggestions. We have worked through the manuscript and we hope that it is adequate now. Corrections can also be verified by the attached file. Please see the attachment.

1st comment:

Figures 1, 6 and 7 should be removed from the manuscript. Figures do not add anything to the article, they only add to the volume of the manuscript, which is already very large.

Answer:

We concur with the reviewer and sincerely appreciate your astute observation. The figures have been removed.

2nd comment:

Line 320. The sentence is wrong. Methanation is a chemical reaction that converts COx to methane. The process described by the authors is anaerobic digestion. This process produces methane, but also CO2. This part of the manuscript needs to be corrected. Answer:

We concur with the reviewer and appreciate your astute observation. The text has been corrected. The sentence excerpt is from the corrected text:

“In contrast, to aerobic processes, which produce CO2, the anaerobic process results in the generation of both CO2 and CH4 [25, 26].”

3rd comment:

The use of SEM to assess biodegradation efficiency is debatable and questionable. It is not an objective method. The operator can always find an image that confirms his research hypothesis. I propose to remove information on this subject from the work (including Fig.5, which does not prove anything), or only to indicate that such a method also exists. Moreover, the work is very extensive.

Answer:

We appreciate the reviewer's suggestion. We prefer to retain the Figure, as its purpose is to emphasize the utilization of SEM as a tool for evaluating the degradation process, specifically by illustrating macroscopic changes and potential erosion effects in polymeric films. Moreover, these images are original and authored by the authors of this paper, effectively depicting the alterations discussed in the text and attributing citations to the journal Macromol, rather than the authors of Figures from other publications. Furthermore, this illustration has been commonly referenced in other review and research articles, as demonstrated by the following examples:

Gazvoda, L., Višić, B., Spreitzer, M., & Vukomanović, M. (2021). Hydrophilicity Affecting the Enzyme-Driven Degradation of Piezoelectric Poly-l-lactic Films. Polymers, 13(11), 1719. doi:10.3390/polym13111719

Scaffaro, Maio, Sutera, Gulino, & Morreale. (2019). Degradation and Recycling of Films Based on Biodegradable Polymers: A Short Review. Polymers, 11(4), 651. doi:10.3390/polym11040651

Nevoralová, M., Koutný, M., Ujčić, A., Starý, Z., Šerá, J., Vlková, H., Kruliš, Z. (2020). Structure Characterization and Biodegradation Rate of Poly(ε-caprolactone)/Starch Blends. Frontiers in Materials, 7. doi:10.3389/fmats.2020.00141

Eich, A., Mildenberger, T., Laforsch, C., & Weber, M. (2015). Biofilm and Diatom Succession on Polyethylene (PE) and Biodegradable Plastic Bags in Two Marine Habitats: Early Signs of Degradation in the Pelagic and Benthic Zone? PLOS ONE, 10(9), e0137201. doi:10.1371/journal.pone.0137201

Reviewer 2 Report

The review with the title of Biodegradation of polymers: stages, measurement, standards, and prospects conclude by highlighting strategies and perspectives, such as recycling and circularity, to make polymers more environmentally friendly and to reduce the need for throwing away plastics. The work seems to be valuable due to its novelty and high-quality content. It provides important insights into the expanding world of sustainable and biodegradable bioplastics, including issues such as greenwashing and the need for better standards and evaluation methods. Such a paper can be a valuable resource for researchers, policymakers, and the general public who are interested in sustainable materials and reducing plastic pollution.

Author Response

Reviewer 2:

1st comment

The review with the title of Biodegradation of polymers: stages, measurement, standards, and prospects conclude by highlighting strategies and perspectives, such as recycling and circularity, to make polymers more environmentally friendly and to reduce the need for throwing away plastics. The work seems to be valuable due to its novelty and high-quality content. It provides important insights into the expanding world of sustainable and biodegradable bioplastics, including issues such as greenwashing and the need for better standards and evaluation methods. Such a paper can be a valuable resource for researchers, policymakers, and the general public who are interested in sustainable materials and reducing plastic pollution.

Answer: We express our sincere gratitude to the reviewer for the insightful and professional feedback on our manuscript.

Reviewer 3 Report

The submitted work in its current form doesn't appears to be adequate to the reviewer. Aside major issues with the written form and the clarity of the representations the represented content is not significantly new or original to the field in order to commit it at as review article. It is a very basic summary of some aspects in the field of biopolymers and biodegradation. It seems to be very difficult to the reviewer to cover such a broad topic within one single review article without choosing some aspects within the field to highlight challenges, approaches or current opinions in the public, industrial and political field. The authors might reconsider changing the scope and the content of the review to a more specific question in the field of biodegradation of polymers.

Author Response

Reviewer 3:

Thank you very much for your consideration. We have worked through the manuscript and we hope that it is adequate now. Corrections can also be verified by the attached file. Please see the attachment.

1st comment

The submitted work in its current form doesn't appears to be adequate to the reviewer. Aside major issues with the written form and the clarity of the representations the represented content is not significantly new or original to the field in order to commit it at as review article. It is a very basic summary of some aspects in the field of biopolymers and biodegradation. It seems to be very difficult to the reviewer to cover such a broad topic within one single review article without choosing some aspects within the field to highlight challenges, approaches or current opinions in the public, industrial and political field. The authors might reconsider changing the scope and the content of the review to a more specific question in the field of biodegradation of polymers.

Answer:

We extend our sincere appreciation to the esteemed reviewer for their invaluable feedback, and we would like to highlight several notable strengths of our article. Our study rigorously delves into the intricate mechanisms of polymer degradation, encompassing a comprehensive classification of degradation rates based on intrinsic factors such as binding type and hydrophobicity. Furthermore, we diligently address and rectify a common error in the existing literature pertaining to degradation events and the accurate conceptualization of decomposition temperature.

In addition, we provide a concise yet comprehensive overview of the various stages of biodegradation, supplemented by an additional schematic diagram that eloquently illustrates each step. Moreover, we elucidate how an array of characterization tools, in alignment with updated international standardized norms, collectively contribute to a nuanced understanding of the (bio)degradation process, thereby empowering readers to make informed decisions. Moreover, the manuscript has been meticulously reviewed by a native English-speaking reviewer, and necessary corrections have been implemented to ensure the accuracy and quality of the language used throughout the article.

Furthermore, we expound upon the societal and political approaches towards advancing the widespread implementation of biodegradable and sustainable materials, thereby making meaningful contributions to the ever-evolving knowledge frontier in this domain. Despite the breadth of our article, we are confident that the unique perspectives presented herein will garner valuable citations and recognition from the scientific community.

In summary, our article provides a robust analysis of polymer degradation, presents updated information on state-of-the-art characterization techniques, and delves into the societal and political implications of biodegradable and sustainable materials. We wholeheartedly hope that our work is well-received and contributes significantly to the advancement of knowledge in this field.

Reviewer 4 Report

This manuscript is well organized. I think that it can be accepted after minor revision.

Some comments:

1. The "Abstract"  was too general, some important information and results need be provided.

2. In "2. Biodegradable Polymers and Stages of Biodegradation", the related enzymes and bacteria need be summarized in one Table. And the related mechanisms are required to be discussed.

3. "6. Final considerations" was too short. Please include more information what can be done for the future studies.

Author Response

Dear reviewer,

Thank you very much for your consideration and suggestions. We have worked through the manuscript and we hope that it is adequate now.

1st comment

The "Abstract"  was too general, some important information and results need be provided.

Answer: We appreciate your feedback and understand your concern about the level of detail provided in the abstract. However, as a review article, the abstract is intended to provide a general overview of the topics covered in the manuscript, avoiding the loss of important content sections. This makes the reader understand the topics discussed and easily find the topics of interest. We hope you accept the current standard.

2nd comment

In "2. Biodegradable Polymers and Stages of Biodegradation", the related enzymes and bacteria need be summarized in one Table. And the related mechanisms are required to be discussed.

Answer: We express our sincere gratitude to the reviewer for the insightful and professional feedback on our manuscript. As per the reviewer's suggestion, we have created a table that highlights the main enzymes and bacteria involved in biodegradation of polymers, along with their respective mechanisms and modes of action. Here is the inserted Table.

Table 1. Enzymes and bacteria involved in biodegradation of polymers.

Type of Enzyme/Bacteria

Polymer Type

Biodegradation Mechanism

Mode of Action and Mechanisms

Proteases

Proteins

Hydrolysis

Catalyze the cleavage of peptide bonds in proteins, breaking them down into smaller peptides and eventually amino acids.

Lipases

Lipids

Hydrolysis

Break down ester bonds in lipids, producing free fatty acids and glycerol.

Amylases

Starch

Hydrolysis

Break down the α-1,4-glycosidic bonds in starch, producing glucose.

Cellulases

Cellulose

Hydrolysis

Break down the β-1,4-glycosidic bonds in cellulose, producing glucose.

Chitinases

Chitin

Hydrolysis

Break down the β-1,4-glycosidic bonds in chitin, producing N-acetylglucosamine.

Laccases

Lignin

Oxidation

Oxidize the phenolic and non-phenolic structures in lignin, breaking down the polymer into smaller fragments.

Peroxidases

Lignin

Oxidation

Catalyze the oxidation of lignin by hydrogen peroxide or oxygen, breaking it down into smaller fragments.

Kosakonia sp.

Polyethylene

Anaerobic metabolism

Production of extracellular enzymes to break down polyethylene into smaller fragments for cellular uptake and utilization as carbon and energy sources.

Aspergillus sp.

Various

Aerobic metabolism

Produce reactive oxygen species and a range of extracellular enzymes, e.g., cellulases, hemicellulases, and ligninases.

The data in the table were constructed based on references [18-20].

3rd comment

Final considerations" was too short. Please include more information what can be done for the future studies.

Answer: We concur with the reviewer and sincerely appreciate your astute observation. More information about what can be done for future studies has been added in the final considerations. Here are the inserted texts:

This review provides an overview of the main stages and mechanisms of polymer biodegradation. Despite the increasing interest and progress in biodegradable polymers, there are still several limitations that need to be addressed to promote their widespread adoption as a sustainable alternative to conventional plastics. The following points outline important areas where further research is necessary to make progress in the field of biodegradable polymers.

(I) Investigation of the effects of additives and contaminants on biodegradation: The majority of plastics contain additives that can affect their properties, e.g., color, flexibility, and durability. However, these additives can also interfere with the biodegradation process, and in some cases, release toxic substances. Further research could investigate how different types of additives and contaminants affect biodegradability, and develop strategies to remove or mitigate their impact.

(II) Use of microbial consortia and genetic engineering to enhance biodegradation: Microbial consortia are communities of microorganisms that work together to degrade complex compounds. Genetic engineering techniques can be used to modify microorganisms to enhance their biodegradation capabilities. Research in this area could involve identifying new microbial consortia and engineering them to enhance their ability to degrade specific types of polymers.

(III) Advanced materials design: There is an opportunity to develop advanced materials with superior biodegradability and functionality. One approach is to design polymers with built-in degradation mechanisms, such as responsive polymers that break down in response to specific stimuli. Another approach is to develop materials that are composed of multiple types of polymers, each with different degradation rates, which could enable more precise control over the biodegradation process.

(IV) Development of circular economy strategies: In addition to developing new biodegradable polymers, research could focus on strategies for closing the loop on plastics. This could involve developing new recycling technologies that can efficiently recycle a wide range of plastics, as well as exploring alternative uses for waste plastics, such as energy recovery or conversion into other materials.

(V) Synthetic biology: Synthetic biology is a rapidly advancing field that involves the design and construction of new biological systems for specific applications. In the context of polymer biodegradation, synthetic biology could be used to engineer microorganisms with enhanced biodegradation capabilities, or to develop synthetic enzymes that can break down specific types of polymers.

(VI) Synthetic biology for upcycling: In addition to engineering microorganisms to enhance biodegradation capabilities, synthetic biology can also be used to engineer microorganisms to upcycle plastics into new materials with higher value. For example, using bacteria to convert plastics into renewable chemicals that can be used to make new plastics or other materials.

(VII) Nanotechnology: Nanotechnology involves the manipulation of materials at the nanoscale to create new properties and functionality. In the context of polymer biodegradation, nanotechnology could be used to develop new materials with enhanced biodegradability or to create new methods for breaking down plastics into their constituent parts. For example, developing nanocatalysts that can accelerate the breakdown of plastics and promote mineralization.

(VIII) Waste-to-energy technologies: In the context of biodegradable polymers, waste-to-energy technologies could be used to convert biodegradable plastics or waste materials into energy, such as biogas or biofuel, through anaerobic digestion or other processes. This could provide a more sustainable and efficient way to dispose of biodegradable plastics than landfilling or incineration.

Reviewer 5 Report

Comments to authors

Title: Biodegradation of Polymers: Stages, Measurement, Standards and Prospects (macromol-2299098)

The fundamental question addressed in this manuscript is: The main stages and mechanisms of polymer biodegradation, the current steps and mechanisms of polymer biodegradation are elucidated. Finally, current strategies that enable the measurement of biodegradation processes and the concept of recycling are summarized to make biodegradable polymers more environmentally friendly and sustainable. Overall, the paper's scientific value would be good enough to be published in macromol after the authors' minor revision of the manuscript considering the following comments

1.   Please re-define “bioplastic” in the introduction and definitions.

2.   In this sentence "One of the main causes is directed mass production and increased use of plastics that are still largely produced from fossil sources and present a non-biodegradable behavior". However, Poly (butyleneadipate-co-terephthalate) (PBAT) is a biodegradable polymer that made from fossil sources, contradicting the non-biodegradable behavior described by the authors.

3.   Biodegradable Plastics, the follow references were referred, which may give some help and insights about the revisions. Such as Carbohydrate Polymers, 2023, 304, 120511; Journal of Cleaner Production, 2020, 258:120536; Journal of King Saud University - Science, 2021, 33(6):101538.

4.   In Prospects. The authors were asked to summarize how well the public currently understands biodegradability and to analyze the reasons for this phenomenon.

5.   Please cite references from the last decade, such as references 13, 21, 23, 26, …….

6.   The reference format is not uniform, such as references 17~25, ……. Please re-edit all of the references.

Author Response

Dear reviewer,

We express our gratitude for your thoughtful consideration and valuable suggestions. Following the receipt of your feedback, we have thoroughly revised the manuscript and trust that it now meets the requisite standards.

1st comment:

Please re-define “bioplastic” in the introduction and definitions.

Answer:

We are grateful for your observation, however this definition has already been corrected at the editor's suggestion. We believe that this current definition is concise and accurate, preventing the spread of a misconception.

2nd comment:

In this sentence "One of the main causes is directed mass production and increased use of plastics that are still largely produced from fossil sources and present a non-biodegradable behavior". However, Poly (butyleneadipate-co-terephthalate) (PBAT) is a biodegradable polymer that made from fossil sources, contradicting the non-biodegradable behavior described by the authors.

Answer:

Thank you for your careful and insightful observation. We agree with the reviewer and corrected the sentence. The meaning of the phrase is to indicate that those responsible for the environmental impact are plastics produced from fossil sources and that they are also non-biodegradable, that is, excluding polycaprolactone (PCL) and poly (butyleneadipate-co-terephthalate). Please see the excerpts text.

“One of the main causes is the mass production and increased use of plastics that are still largely produced from fossil sources and which exhibit non-biodegradable behavior [2].”

3rd comment:

Biodegradable Plastics, the follow references were referred, which may give some help and insights about the revisions. Such as Carbohydrate Polymers, 2023, 304, 120511; Journal of Cleaner Production, 2020, 258:120536; Journal of King Saud University - Science, 2021, 33(6):101538.

Answer:

Thank you to the reviewer for the valuable feedback. We have carefully considered your suggestion and the articles have been added, as well as some enhancements throughout the article to address your concerns.

4rth comment:

In Prospects. The authors were asked to summarize how well the public currently understands biodegradability and to analyze the reasons for this phenomenon.

Answer:

We appreciate the opportunity to clarify our article based on your suggestions. In response to your comments, we have elaborated on how well the public understands biodegradability, the reasons behind this lack of understanding, and presented strategies for raising awareness about biodegradable polymers. Additionally, we have included additional topics in item 4 of the article to highlight the main evidence for biodegradation, contributing to the identification of the biodegradation process. Please see the excerpts below for further details.

In turn, the understanding of biodegradability of polymers among the general public is currently low due to several reasons, including lack of education and awareness, complex scientific terminology, and limited access to accurate information. To boost understanding, a disruptive way would be to use innovative communication strategies that simplify complex scientific concepts and make them easily accessible to the public. To address this issue and promote greater public awareness of polymer biodegradability, innovative communication strategies could be employed, such as those presented below.

  • Public education campaigns: Develop a targeted public education campaign focused on promoting awareness of biodegradable polymers, their benefits, and their proper disposal. This could include media outreach, social media campaigns, and public events.
  • Collaborations with industries: Partner with industries that use polymers to promote the use of biodegradable alternatives and educate consumers on their benefits.
  • Improved labeling: Develop standardized labeling for biodegradable polymers that is easy for consumers to understand and includes information on proper disposal.
  • School curriculum: Incorporate the science of biodegradable polymers into school curriculum, starting at a young age. This can help to create a more informed and environmentally conscious future generation.

5th comment:

Please cite references from the last decade, such as references 13, 21, 23, 26, …….

Answer:

We express our sincere gratitude to the reviewer for the professional feedback on our manuscript. Nevertheless, we recognize the value of citing established and renowned articles. We believe it is important to give credit to authors who have presented disruptive ideas and concepts. In this review article, 83% of the references in this review article are from the last decade. Thus, we aim to honor the creators of long-standing proposals as well as those who continue to push the boundaries with new concepts.

6th comment:

The reference format is not uniform, such as references 17~25, ……. Please re-edit all of the references.

Answer:

Thank you for your attentive and careful observation. All references have been revised again.

Round 2

Reviewer 3 Report

While the authors have improved their manuscript in terms of language, my prior stated major objections remain. 

lines 72-73: There is no lack of scientific evidence concerning the real biodegradability. The authors reference in their manuscript a significant amount of scientific literature covering this topic

lines 111: Why do the authors define a specific definition and what is the reason for their choice? In how far is this definition "fair"?

Figure 2: What is the classification based on? Experimental results or calculations? Literature references or own work? Which conditions?

The text seems to adequately reflect the complexity of the hydrolytic reaction kinetics (hydrophobicity, crystallinity, bulk vs solution, etc.) while the figure as shown (see comments above) is too general.

Figure 3: Redundant representations and misleading or wrong caption.

Figure 5: Missing caption

Additional remark from the reviewer to the authors:

The manuscript from paragraph 3 on, seems to be a good and relevenat representation of the technical aspects on the degradability of polymers. I can imagine a well written review by expanding and combining to aspects of sustainability as already mentioned in the introduction (and in my humble opinion another controversial subject in the field of industrial polymers).

Author Response

Dear Reviewer,

Thank you very much for your consideration and suggestions. We have worked through the manuscript and incorporated additional enhancements, including a new section on future research and other improvements, as recommended by others reviewers. This section highlights several disruptive approaches that could pave the way for expanding the current frontiers of biodegradable polymers. We hope that it is adequate now.

1st comment

Lines 72-73: There is no lack of scientific evidence concerning the real biodegradability. The authors reference in their manuscript a significant amount of scientific literature covering this topic.

Answer:

We appreciate the careful review and agree with the reviewer. The focus of this sentence is to discuss the current state of biodegradation of bioplastics, and to highlight the fact that commonly used biodegradation tests only evaluate the initial phases of biodegradation, i.e., biodeterioration and biodisintegration. Additionally, we aim to draw attention to the issue of evaluating biodegradability in blends and polymer composites, which is often overlooked and the biodegradability characteristic is scarcely evaluated despite claims of biodegradability. In this way, it is possible to link this problem with the definition of biodegradability that appears in consecutive lines.

The sentence excerpt is from the corrected text:

“However, there is still a lack of scientific evidence concerning the real biodegradability and sustainability of these novel plastics, especially when structured in the form of composites and/or blends [7].”

2nd comment:

Lines 111: Why do the authors define a specific definition and what is the reason for their choice? In how far is this definition "fair"?

Answer:

We sincerely appreciate your astute observation. We defined polymer degradation in order to provide a precise and unambiguous understanding of the term for readers. This definition is crucial for readers to comprehend the mechanisms and consequences of polymer degradation that are discussed in subsequent sections. As a review article, it is essential to accurately present the main definitions, bringing citations to the article and journal.

We agree with the reviewer's point and have expanded the definition to encompass the full nature of polymer degradation, acknowledging the various types of reactions that can lead to it. This makes the definition more fair and equitable across different approaches. We hope that this revision will meet the reviewer's standards.

The sentence excerpt is from the corrected text:

“Polymer degradation refers to any chemical, physical, or biochemical reaction that involves breaking covalent bonds in the backbone of the polymer, resulting in an irre-versible change in its properties due to alterations in the chemical structure and the reduc-tion of molecular weight.”

3rd comment:

Figure 2: What is the classification based on? Experimental results or calculations? Literature references or own work? Which conditions?

The text seems to adequately reflect the complexity of the hydrolytic reaction kinetics (hydrophobicity, crystallinity, bulk vs solution, etc.) while the figure as shown (see comments above) is too general.

Answer:

Thanks to the reviewer for the thoughtful comment. The purpose of this figure is to provide a general classification of the main characteristics that influence the degradation of polymers when exposed to humidity. The classification is based on well-established findings from the literature, with experimental results, drawing upon highly regarded books and articles. Notably, the citation book [13], "Polymer Science" which includes 1,304 citations, was instrumental in enabling this classification. Additional scientific articles [12, 14-16] also provide substantial evidence to support the claims made in this study. Therefore, this classification was created by the authors through the collection of information based on the literature, and citations were made in the main body of the text to acknowledge the sources used.

4rth comment:

Figure 3: Redundant representations and misleading or wrong caption.

Answer:

We appreciate the reviewer's comment, and we believe that Figure 3 is crucial to illustrate two of the main reactions that occur in the degradation of polymers and to present the stages of biodegradation discussed in the article. Specifically, Figure 3c provides a concise depiction of the factors involved in biodegradation. We are confident that including Figure 3 in the manuscript will enhance readers' understanding and engagement with our work.

5th comment:

Figure 5: Missing caption.

Answer:

We appreciate the reviewer's comment, but the current manuscript does not have Figure 5.

Round 3

Reviewer 3 Report

My major concerns remain. While the authors are improving the article in each iteration, I question the relevance of this review to the field and leave this decision, with all respect, to the editor.